# On the Alignment between Fairness and Accuracy: from the Perspective of Adversarial Robustness

Junyi Chai [1]    Taeuk Jang [1]    Jing Gao [1]    Xiaoqian Wang [1]

## Abstract

While numerous work has been proposed to address fairness in machine learning, existing methods do not guarantee fair predictions under imperceptible feature perturbation, and a seemingly fair model can suffer from large group-wise disparities under such perturbation. Moreover, while adversarial training has been shown to be reliable in improving a model's robustness to defend against adversarial feature perturbation that deteriorates accuracy, it has not been properly studied in the context of adversarial perturbation against fairness. To tackle these challenges, in this paper, we study the problem of adversarial attack and adversarial robustness w.r.t. two terms: fairness and accuracy. From the *adversarial attack* perspective, we propose a unified structure for adversarial attacks against fairness which brings together common notions in group fairness, and we theoretically prove the equivalence of adversarial attacks against different fairness notions. Further, we derive the connections between adversarial attacks against fairness and those against accuracy. From the *adversarial robustness* perspective, we theoretically align robustness to adversarial attacks against fairness and accuracy, where robustness w.r.t. one term enhances robustness w.r.t. the other term. Our study suggests a novel way to unify adversarial training w.r.t. fairness and accuracy, and experiments show our proposed method achieves better robustness w.r.t. both terms.

## 1. Introduction

As machine learning systems have been increasingly applied in high-stake fields, it is imperative that machine learning

---

[1]Elmore Family School of Electrical and Computer Engineering, Purdue University. Correspondence to: Xiaoqian Wang <joywang@purdue.edu>.

*Proceedings of the 42$^{nd}$ International Conference on Machine Learning*, Vancouver, Canada. PMLR 267, 2025. Copyright 2025 by the author(s).

models do not reflect real-world discrimination. However, machine learning models have shown biased predictions against disadvantaged groups on several real-world tasks (Larson et al., 2016; Dressel & Farid, 2018; Mehrabi et al., 2021a). In order to improve fairness and reduce discrimination of machine learning systems, a variety of work has been proposed to quantify and rectify bias (Hardt et al., 2016; Kleinberg et al., 2016; Mitchell et al., 2018). Despite the emerging interest in fairness, fairness depreciation in the context of adversarial perturbation and the corresponding defense techniques have not been adequately discussed.

Previous work has demonstrated that by applying small magnitude of adversarial perturbations to input features, the performance (classification accuracy) of machine learning models can be severely deteriorated (Goodfellow et al., 2014; Madry et al., 2017). Realizations of such perturbations are generally referred to as the adversarial attack and have been widely discussed in current literature (Croce et al., 2020; Bai et al., 2021). Being the complementary of adversarial attack, the adversarial robustness provides reliable quantification regarding the statistical properties of a machine learning model when subjected to such adversarial perturbations, particularly concerning accuracy. Methods that focus on improving adversarial robustness are referred to as adversarial training (Chakraborty et al., 2018; Wong et al., 2020; Sriramanan et al., 2021). However, such discussion becomes troublesome when extended to the statistical parity of a model, i.e., the adversarial robustness in the context of fairness. Regarding fairness, the adversarial perturbations aim at exacerbating disparities between sensitive groups, typically characterized by the group fairness notions, including demographic parity (DP) and equalized odds (EOd). Owing to the variations in the statistical characteristics of fairness and accuracy, such perturbations may not necessarily align with the adversarial attack. As shown in Tab. 1, while the adversarial attack effectively approximates the worst-case perturbations against the accuracy, it fails to provide appropriate estimations regarding the worst-case perturbations against fairness measures, especially in terms of EOd. Under a successful adversarial attack, the difference in group-wise classification errors vanishes, leading to zero violation in EOd. Consequently, a successful adversarial attack does not necessarily ensure fairness depreciation, and

vice versa.

| $\epsilon$ | Acc | DP | EOd |
|---|---|---|---|
| 0 | 84.25±1.17% | 14.34±1.65% | 28.45±2.27% |
| 0.3 | 0 | 15.13±0.17% | 0 |

*Table 1.* Comparison of changes in statistical measures (Acc for accuracy, DP and EOd for fairness) under adversarial attacks on Adult dataset. Lower DP and EOd indicates smaller disparities between sensitive groups.

As discussed in (Solans et al., 2020; Mehrabi et al., 2021b; Chhabra et al., 2022; Kang et al., 2023), similar to accuracy, fairness can also be targeted by malicious adversarials, leading to biased outcomes against certain demographics. While previous work has been focused on the poisoning attack against fairness measures, the adversarial attack and adversarial robustness in the context of fairness has been rarely discussed. Given that current adversarial attack techniques are not readily applicable to fairness depreciation through adversarial perturbation, there is a pressing necessity to develop tailored methods for formulating the adversarial attacks, as well as the adversarial robustness, within the context of fairness.

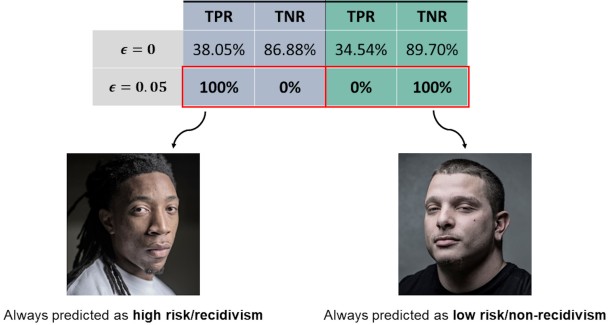

*Figure 1.* Demonstration of adversarial attacks against fairness on COMPAS dataset (Larson et al., 2016), where the statically fair classifier is obtained by in-processing (Wang et al., 2022). Under a small perturbation level $\epsilon = 0.05$ ($\leqslant 1.5\%$ of input feature's norm), the disparities in true positive rates (TPR) and true negative rates (TNR) between white and black people increase sharply to 100%, and the outcomes of classifier become solely dependent of sensitive information, leading to destructive social injustice against the disadvantaged group (where all the black individuals will be considered as of high risk in this demonstration). Fairness shall not be considered as a static measure, and a classifier with small fairness gaps can show large disparities under fairness attacks. Image credit: (Angwin et al., 2016).

For simplicity of discussion, with slight abuse of phraseology we define the adversarial perturbations against accuracy as the **accuracy attack**, i.e., the imperceptible perturbation to deteriorate accuracy, and **accuracy adversarial samples**

as samples that are adversarially perturbed by the accuracy attack. We leave the mathematical formulations in Sec. 3.1. We define such robustness to the accuracy attack as **accuracy robustness**, i.e., a model's ability to resist adversarial perturbation by an **accuracy attack** and remain same predictions on clean and accuracy adversarial samples. Similar to those of accuracy, we define the **fairness attack** as the imperceptible perturbation to deteriorate fairness, and **fairness adversarial samples** as samples that are adversarially perturbed by the fairness attack. We show the mathematical formulations in Sec. 3.2. Specifically, while accuracy attacks aim at exacerbating the classification error, fairness attacks try to deteriorate group-wise parity between different sensitive groups, leading to varied perturbations up to each individual. Work including (Solans et al., 2020; Mehrabi et al., 2021b) first proposed to generate fairness adversarial samples taking into account fairness objectives to perturb the training process and to exacerbate bias on clean testing data. However, the detailed mechanism of fairness attacks has not yet been properly discussed, as it remains unclear how the instance-wise perturbations work, and the relationship between fairness attacks and accuracy attacks remains ambiguous. Moreover, the adversarial robustness regarding fairness attacks, as well as its connection with accuracy robustness, has been overlooked in current literature.

Just as a model optimized for accuracy in training may not be robust against an accuracy attack, similarly, a fair model trained for static fairness may not inherently possess fairness robustness against fairness attacks. Here we similarly define **fairness robustness** as a model's ability to resist adversarial perturbation by a **fairness attack** and remain same predictions on clean and fairness adversarial samples. As shown in Fig. 1, fairness can be volatile under adversarial perturbations, where a small degree of perturbation can lead to significant variations in group-wise disparities, and enforcing fairness alone during training does not necessarily lead to improvement in robustness against fairness attacks. In fairness literature, although adversarial training has been widely discussed, most of them have been focused on applying adversarial training as a means to unlearn the impact of sensitive attributes to achieve static fairness (Madras et al., 2018; Creager et al., 2019). Chhabra et al. (2022) first propose a defense framework for adversarial perturbation against fairness; however, such perturbation is targeted against sensitive information, rather than input features.

In this work, we propose a general framework for fairness attacks, where we show impacts of fairness attacks up to each individual under different notions, as well as the connections between these notions regarding gradient-based attacks. Based on this unified framework, we discuss the relationship between fairness attacks and accuracy attacks. Furthermore, we show that despite the discrepancies in adversarial perturbations between the fairness attack and the

accuracy attack for certain samples, fairness robustness and accuracy robustness do not necessarily conflict with each other. Based on the spatial proximity between such samples and samples where the fairness attack and the accuracy attack acts in the same direction, we show theoretically how fairness robustness and accuracy robustness can benefit from each other, i.e., the alignment between fairness robustness and accuracy robustness. Our theoretical results suggest a novel defense framework, *fair adversarial training*, which incorporates fair classification with adversarial training so as to improve fairness robustness. We summarize our contribution as follows:

- We propose a unified framework for fairness attacks, which brings together different notions in group fairness.

- We theoretically demonstrate the alignment between fairness robustness and accuracy robustness, and we propose a novel defense framework, *fair adversarial training*, which incorporates fairness robustness with fair classification.

- We empirically validate the superiority of our method under fairness attacks, and the connection between fairness robustness and accuracy robustness on four benchmark datasets.

## 2. Related Work

**Fairness in machine learning**. Fairness has gained much attention in machine learning society. Different notions have been proposed to quantify discrimination of machine learning models, including individual fairness (Lahoti et al., 2019; John et al., 2020; Mukherjee et al., 2020), group fairness (Feldman et al., 2015; Hardt et al., 2016; Zafar et al., 2017) and counterfactual fairness (Kusner et al., 2017). Our work is most closely related with group fairness notions. Works on group fairness generally fall into three categories: preprocessing (Creager et al., 2019; Jiang & Nachum, 2020; Jang & Wang, 2024; Yu et al., 2024; Jung et al., 2025), where the goal is to adjust training distribution to reduce discrimination; in-processing (Zafar et al., 2017; Jung et al., 2021; Roh et al., 2021; Chai & Wang, 2022; Lu et al., 2024), where the goal is to impose fairness constraint during training by reweighing or adding relaxed fairness regularization; and post-processing (Hardt et al., 2016; Jang et al., 2022; Cruz & Hardt, 2023; Ţifrea et al., 2023), where the goal is to adjust the decision threshold in each sensitive group to achieve expected fairness parity. Compared with existing work, we extend the discussion to both static fairness and fairness robustness, emphasizing the fairness notions under adversarial scenarios.

**Adversarial machine learning**. Adversarial training and adversarial attack have been widely studied in trustworthy machine learning. Goodfellow et al. (2014) propose a simple one-step gradient-based attack to adversarially perturb the input features. Madry et al. (2017) extend the one-step attack to an iterative attack strategy and show that iterative strategy is better at finding accuracy adversarial samples. Accordingly, different methods on adversarial defense have been proposed (Shafahi et al., 2019; Wong et al., 2020; Xie et al., 2020; Cui et al., 2021; Jia et al., 2022; Dong & Xu, 2023; Ma et al., 2024) to improve accuracy robustness of a classifier. However, few literature has addressed the adversarial training and adversarial attack against fairness. Some work discusses the problem of fairness poisoning attack during training (Solans et al., 2020; Mehrabi et al., 2021b); however, it is not clear how fairness attacks would influence the predicted soft labels.

**Fairness in adversarial robustness**. Group fairness in the context of adversarial robustness has been less studied in current work. Work including (Nanda et al., 2021; Xu et al., 2021; Ma et al., 2022) argues that adversarial training without proper regularization leads to class-wise disparities in accuracy and robustness. However, group-wise disparities are not considered in these work, and the formulation of disparities by these work is not in accord with notions in group fairness. Recent work studies the poisoning attack against group fairness measures (Solans et al., 2020; Mehrabi et al., 2021b; Zeng et al., 2023); however, these work lacks in-depth discussion regarding the detailed mechanism of the fairness attack and the defense techniques. Specifically, the relationship between the fairness attack and the accuracy attack, as well as that between fairness robustness and accuracy robustness remains unclear. Our work is most related to (Chhabra et al., 2022), where adversarial perturbation against sensitive information and the corresponding defense mechanism are considered. In comparison, our framework considers feature-level perturbation, rather than sensitive-information-level , which provides more general quantification regarding the worst-case perturbations against fairness.

## 3. Problem Definition

### 3.1. Adversarial attack against accuracy

We start by formulating the **accuracy attack**. Denote $x \in \mathbb{R}^n$ as the input feature, $y \in \{0, 1\}$ as the label, and $a \in \{0, 1\}$ as the sensitive attribute. Let $f : \mathbb{R}^n \to [0, 1]$ be the function of classifier and $f(x)$ the predicted soft label, then the objective of accuracy attack for sample $(x, y, a)$ can be formulated as

$$\arg\max_{\epsilon} L_{\text{CE}}(f(x + \epsilon), y), \text{ s.t.} \|\epsilon\| \leq \epsilon_0,$$

where $\|\epsilon\|$ refers to the $L^p$ norm of $\epsilon$ with a general choice of $L^\infty$ norm, and $L_{\text{CE}}$ is the cross-entropy loss. By estimating

the perturbation that maximize $L_{\text{CE}}$, we seek to amplify the difference between the predicted soft label and the ground-truth label, thereby deteriorating accuracy. A common way to obtain accuracy adversarial samples is through projected gradient descent (PGD) attack, where **accuracy adversarial samples** are iteratively updated in each step based on the signed gradient:

$$x^{t+1} = \Pi_{x+S}\left(x^t + \alpha \operatorname{sign}\left(\nabla_x L_{\text{CE}}(x, y)\right)\right),$$

where $\alpha$ is the step size, $S := \{\epsilon, \|\epsilon\| \le \epsilon_0\}$ is the set of allowed perturbation and $\Pi$ is the projection operator that clips the perturbed input into the allowed $\epsilon_0$-ball. PGD attack has been shown to be effective in finding adversarial samples compared with one-step adversarial attack (Madry et al., 2017).

### 3.2. Adversarial attack against fairness

Similar to that of accuracy attack, we formulate the **fairness attack** as follows:

$$\arg\max_\epsilon L(f(x + \epsilon), a, y), \text{ s.t.} \|\epsilon\| \le \epsilon_0,$$

where $L$ is some relaxed fairness constraint. By approximating the worst-case perturbation regarding fairness attack, we seek to amplify the difference in predicted soft labels across different sensitive groups. We consider two widely adopted group fairness notions: demographic parity (DP) and equalized odds (EOd). For a testing set $\mathbb{S} = \{(x_i, y_i, a_i), 1 \le i \le N\}$, denote $\mathbb{S}_{jk} := \{x_i | y_i = j, a_i = k\}$, and $\mathbb{S}_{.k} := \{x_i | a_i = k\}$. The objective can be formulated as relaxations of fairness notions (Wang et al., 2022):

$$L_{\text{DP}} = \left| \sum_{x_i \in \mathbb{S}_{.1}} \frac{f(x_i)}{|\mathbb{S}_{.1}|} - \sum_{x_i \in \mathbb{S}_{.0}} \frac{f(x_i)}{|\mathbb{S}_{.0}|} \right|,$$

$$L_{\text{EOd}} = \sum_y \left| \sum_{x_i \in \mathbb{S}_{y0}} \frac{f(x_i)}{|\mathbb{S}_{y0}|} - \sum_{x_i \in \mathbb{S}_{y1}} \frac{f(x_i)}{|\mathbb{S}_{y1}|} \right|. \tag{1}$$

And **fairness adversarial samples** can be obtained analogous to the accuracy attack via PGD attack:

$$x^{t+1} = \Pi_{x+S}\left(x^t + \alpha \operatorname{sign}\left(\nabla_x L(x, a, y)\right)\right).$$

We summarize the white-box threat model as follows: The adversary has full access to the target model's parameters. Adversarial manipulations are applied at the input level, subject to a maximum allowable perturbation. Fairness attacks aim to maximize group-wise disparities on the testing data, measured by metrics such as DP and EOd. In contrast, accuracy attacks seek to maximize classification error on the testing data.

## 4. Connection between Adversarial Attacks

Before delving into the discussion, we first clarify the notations in Tab. 2. Without loss of generality, we assume $\mathbb{S}_{.1}$ **the advantaged group** with higher average positive prediction. We refer to fairness attacks targeting DP and EOd as DP attack and EOd attack, respectively.

### 4.1. Connection between fairness attacks

We now discuss the detailed relationship between DP attack and EOd attack. The following corollary states the compatibility of the two objectives:

**Corollary 4.1.** *The adversarial objective of EOd attack is lower-bounded by that of DP attack up to multiplicative constants.*

We defer the proof to appendix. Corollary 4.1 shows the connection between adversarial attacks against different group fairness notions, where these attacks perturb the predicted soft labels against sensitive attributes. A successful DP attack also leads to a successful EOd attack, while the opposite does not necessarily hold true. We will focus on **DP attack** for the rest of this paper. In the following context, we refer to DP attack as fairness attack unless otherwise specified. For a given sample $(x_j, y_j, a_j)$ in the advantaged group, we can rewrite $L_{\text{DP}}$ in (1) as:

$$\begin{aligned} L_{\text{DP}} &= \left| \sum_{x_i \in \mathbb{S}_{.1}} \frac{f(x_i)}{|\mathbb{S}_{.1}|} - \sum_{x_i \in \mathbb{S}_{.0}} \frac{f(x_i)}{|\mathbb{S}_{.0}|} \right| \\ &= \frac{f(x_j)}{|\mathbb{S}_{.a_j}|} + \sum_{x_i \in \mathbb{S}_{.a_j} \backslash \{x_j\}} \frac{f(x_i)}{|\mathbb{S}_{.a_j}|} - \sum_{x_i \in \mathbb{S}_{.\hat{a}_j}} \frac{f(x_i)}{|\mathbb{S}_{.\hat{a}_j}|} \\ &= \frac{f(x_j)}{|\mathbb{S}_{.a_j}|} + C_j, \end{aligned} \tag{2}$$

where $\hat{a}_j = |1 - a_j|$ and $C_j$ is a constant w.r.t. $x_j$ since it does not affect $\frac{\partial L_{\text{DP}}}{\partial x_j}$. This shows that the DP attack is expected to *maximize* the prediction in the advantaged group.

Similarly, for a sample $(x_k, y_k, a_k)$ in the disadvantaged group, we have:

$$L_{\text{DP}} = \left| \sum_{x_i \in \mathbb{S}_{.1}} \frac{f(x_i)}{|\mathbb{S}_{.1}|} - \sum_{x_i \in \mathbb{S}_{.0}} \frac{f(x_i)}{|\mathbb{S}_{.0}|} \right| = -\frac{f(x_k)}{|\mathbb{S}_{.a_k}|} + C_k, \tag{3}$$

where $C_k$ is a constant w.r.t. $x_k$ thus does not affect $\frac{\partial L_{\text{DP}}}{\partial x_k}$. (3) shows that the DP attack in the disadvantaged group is contrary to that of advantaged group, where the predictions are expected to be *minimized*.

### 4.2. Connection between the fairness attack and the accuracy attack

We move on to discuss the connection between the fairness attack and the accuracy attack. The following corollary

| | |
|---|---|
| $x_{\mathrm{sub},a}$ | The clean samples in the clean subgroup. |
| $x_{\mathrm{sub},a}^{\mathrm{obj}}$ | The adversarial sample(s) obtained after the attack type obj. |
| $x_{\mathrm{sub},a}^{t,\mathrm{obj}}$ | Adversarial sample(s) generated from the clean subgroup $\{\mathrm{sub},a\}$ at $t$-th iteration targeting attack type $\mathrm{obj} \in \{\mathrm{DP}, \mathrm{EOd}, \mathrm{Acc}\}$. |
| $\mathbb{S}_{\mathrm{sub},a}^{t,\mathrm{obj}}$ | The set of samples in the clean subgroup. |
| $p_{\mathrm{sub},a}^{\mathrm{obj}}$ | The distribution of predicted soft labels in the clean subgroup $\{\mathrm{sub},a\}$ after the attack type obj. |
| $p_{\mathrm{sub},a}$ | The distribution of predicted soft labels in the clean subgroup $\{\mathrm{sub},a\}$ without attack. |

*Table 2.* Summarization of notations.

shows the connection between the fairness attack and the accuracy attack:

**Corollary 4.2.** *The fairness attack and the accuracy attack operate in the same direction regarding true negative (TN) and false positive (FP) samples in the advantaged group and true positive (TP) and false negative (FN) samples in the disadvantaged group.*

We defer the detailed proof to appendix. Notably, the fairness attack and the accuracy attack behave in the opposite direction for the remaining sets of samples (i.e., TP and FN samples in the advantaged group, and TN and FP samples in the disadvantaged group). Specifically, for the two subgroups $\mathbb{S}_{\mathrm{TP},1}$ and $\mathbb{S}_{\mathrm{TN},0}$, the fairness attack aims at maximizing their predicted soft labels as in (2) and (3), respectively. This results in maximizing the predicted soft labels for $\mathbb{S}_{\mathrm{TP},1}$ and minimizing the predicted soft labels for $\mathbb{S}_{\mathrm{TN},0}$. Whereas the accuracy attack seeks to minimize the predicted soft labels for $\mathbb{S}_{\mathrm{TP},1}$ and maximize the predicted soft labels for $\mathbb{S}_{\mathrm{TN},0}$.

Likewise, for the subgroups $\mathbb{S}_{\mathrm{FN},1}$ and $\mathbb{S}_{\mathrm{FP},0}$, the fairness attack tries to 'correct' the predicted soft labels such that the adversarial predictions align with the ground-truth labels. In contrast, the accuracy attack is designed to exacerbate the error within these subgroups. We summarize the connection between the fairness attack and the accuracy attack on various subgroups in Table 3.

| Sensitive Group | Same Direction | Inverse direction |
|---|---|---|
| Disadvantaged | $\mathbb{S}_{\mathrm{TP},0}, \mathbb{S}_{\mathrm{FN},0}$ | $\mathbb{S}_{\mathrm{TN},0}, \mathbb{S}_{\mathrm{FP},0}$ |
| Advantaged | $\mathbb{S}_{\mathrm{TN},1}, \mathbb{S}_{\mathrm{FP},1}$ | $\mathbb{S}_{\mathrm{TP},1}, \mathbb{S}_{\mathrm{FN},1}$ |

*Table 3.* Connection between the fairness attack and the accuracy attack regarding samples in different subgroups.

## 5. Alignment between Fairness Robustness and Accuracy Robustness

We now discuss the alignment between fairness robustness and accuracy robustness. According to Table 3, the relationship between fairness robustness and accuracy robustness is straightforward on the four subgroups in 'Same Direction' category. Since the fairness attack and the accuracy attack operate in the same direction for those samples, the

fairness robustness and accuracy robustness also attain alignment on these samples. Consider sample $x_i$ from the 'Same Direction' groups, by Corollary 4.2 we have:

$$f(x_i^{t,\mathrm{Fair}}) = f\left(\Pi_{x_i+S}\left(x_i^{t-1} + \alpha\,\mathrm{sign}\left(\nabla_{x_i} L_{\mathrm{DI}}(x_i, a_i, y_i)\right)\right)\right)$$
$$= f\left(\Pi_{x_i+S}\left(x_i^{t-1} + \alpha\,\mathrm{sign}\left(\nabla_{x_i} L_{\mathrm{CE}}(x_i, y_i)\right)\right)\right).$$

Under same perturbation level $\epsilon$ and same step size $\alpha$ up to $T$ iterations, the fairness attack and the accuracy attack leads to equivalent perturbations in predicted soft labels regarding $x_i$'s in the 'Same Direction' category. Therefore, it is feasible to leverage existing adversarial training tools targeting accuracy robustness to improve fairness robustness for such samples. However, such alignment cannot be directly extended to the four subgroups in the 'Opposite Direction' category. As the fairness attack and the accuracy attack operate in the opposite direction, it is not straightforward whether there exists alignment or misalignment between fairness robustness and accuracy robustness regarding those samples.

Therefore, in the following we focus our discussion on the four 'Opposite Direction' subgroups in Table 3: $\mathbb{S}_{\mathrm{TP},1}$, $\mathbb{S}_{\mathrm{FN},1}$, $\mathbb{S}_{\mathrm{TN},0}$, $\mathbb{S}_{\mathrm{FP},0}$. Under $\epsilon$-level fairness attack with step size $\alpha$ and up to $T$ iterations, we define $D_{\mathrm{sub},a}^{\mathrm{Fair}} := |L_{\mathrm{CE}}(f(x_{\mathrm{sub},a}^{\mathrm{Fair}}), y) - L_{\mathrm{CE}}(f(x_{\mathrm{sub},a}), y)|$ as the change of cross-entropy loss for sample $x_{\mathrm{sub},a}$ and $\delta_{\mathrm{sub},a}^{\mathrm{Fair}} := |f(x_{\mathrm{sub},a}^{\mathrm{DP}}) - f(x_{\mathrm{sub},a})|$ as the change of $f(x_{\mathrm{sub},a})$. Therefore, $D_{\mathrm{sub},a}^{\mathrm{Fair}}$ and $\delta_{\mathrm{sub},a}^{\mathrm{Fair}}$ are related with **fairness robustness**, and smaller $D_{\mathrm{sub},a}^{\mathrm{Fair}}$ and $\delta_{\mathrm{sub},a}^{\mathrm{Fair}}$ indicate better fairness robustness. Likewise, under $\epsilon$-level accuracy attack with step size $\alpha$ and up to $T$ iterations, we define $D_{\mathrm{sub},a}^{\mathrm{Acc}} := |L_{\mathrm{CE}}(f(x_{\mathrm{sub},a}^{\mathrm{Acc}}), y) - L_{\mathrm{CE}}(f(x_{\mathrm{sub},a}), y)|$ as the change of cross-entropy loss for sample $x_{\mathrm{sub},a}$ and $\delta_{\mathrm{sub},a}^{\mathrm{Acc}} := |f(x_{\mathrm{sub},a}^{\mathrm{Acc}}) - f(x_{\mathrm{sub},a})|$ as the change of $f(x_{\mathrm{sub},a})$, and smaller $D_{\mathrm{sub},a}^{\mathrm{Acc}}$ and $\delta_{\mathrm{sub},a}^{\mathrm{Acc}}$ indicates better **accuracy robustness**. Before we state the detailed relationship, we first state the assumptions we need to prove the relationship:

**Assumption 5.1.** The gradient of $f$ w.r.t. input feature $x$ is Lipschitz with constant $K$.

**Assumption 5.2.** The distributions $p_{\mathrm{sub},a}$ are uniformly bounded by constants $M_{\mathrm{sub},a}$.

Under Assumption 5.1 and 5.2, below we discuss the alignment between fairness robustness and accuracy robustness

in two directions, i.e., how fairness/accuracy robustness improves accuracy/fairness robustness. While the assumption of Lipschitz gradient seems a bit strong, it is a widely used assumption for neural network, and it is feasible to estimate the Lipschitz constant (Fazlyab et al., 2019; Shi et al., 2022). Also, the difference in fairness/accuracy robustness as discussed in the Theorem 5.3 and 5.6 are indeed upper-bounded by the Lipschitz constant $K$, and a smaller $K$ indicates better upper-bounds for the difference in robustness, which also suggests better alignment between fairness robustness and accuracy robustness.

### 5.1. From accuracy robustness to fairness robustness

We first derive the guarantee for fairness robustness by accuracy robustness. We will focus on $x_{FN,1}$ and $x_{FP,0}$, as fairness attack regarding $x_{TP,1}$ and $x_{TN,0}$ does not affect fairness, i.e., the predicted labels for $x_{TP,1}$ and $x_{TN,0}$ will remain the same before and after the fairness attack.

**Theorem 5.3.** *Given a classifier $f$, consider $\epsilon$-level fairness attack with step size $\alpha$ and up to $T$ iterations, the difference of fairness robustness between $x_{FN,1}$ and $x_{FN,0}$ is upper-bounded by the accuracy robustness of $x_{FN,0}$ up to an additive and a multiplicative constant:*

$$D_{FN,1}^{Fair} \leq \min_{x_{FN,0} \in \mathbb{S}_{10}} D_{FN,0}^{Acc} + \alpha \sum_{t=1}^{T} G_t,$$

$$G_t = \left[ \frac{\sqrt{n}K d(x_{FN,1}^{t-1,Fair}, x_{FN,0}^{t-1,Fair})}{f(x_{FN,1}^{t-1,Fair})} + \eta_t \delta_{FN,0}^{t-1,Acc} \right],$$

$$\eta_t = \left| \frac{f(x_{FN,0}^{t-1,Fair}) - f(x_{FN,1}^{t-1,Fair})}{f(x_{FN,1}^{t-1,Fair}) f(x_{FN,0}^{t-1,Fair})} \right|.$$

Detailed proof and empirical verification can be found in the appendix. As discussed in Section 4.2, adversarial training w.r.t. accuracy also improves fairness robustness of subgroup $\mathbb{S}_{FN,0}$ while it is unclear for subgroup $\mathbb{S}_{FN,1}$. Thus, we leverage $x_{FN,0}$ to explore robustness guarantee against fairness attack for $x_{FN,1}$. Specifically, for $f'$ under adversarial training w.r.t. accuracy and $f$ under normal training, we have similar upper-bound, except that we now have $\delta_{FN,0}^{'t-1,Acc} \leq \delta_{FN,0}^{t-1,Acc}$, which indicates a tighter upper-bound for $f'$ in Theorem 5.3. For the marginal advantaged FN samples ($x_{FN,1}$) which are more vulnerable under the fairness attack, we have their fairness robustness bounded by marginal disadvantaged FN samples ($x_{FN,0}$), and smaller $\delta_{FN,0}^{Acc}$, or tighter bound indicates better fairness robustness. Similar inequality in Theorem 5.3 also holds for $x_{FP,0}$ and $x_{FP,1}$:

*Remark* 5.4. For $x_{FP,0}$ and $x_{FP,1}$, we have similar inequality regarding the upper-bound of robustness difference:

$$D_{FP,0}^{Fair} \leq \min_{x_{FP,1} \in \mathbb{S}_{01}} D_{FP,1}^{Acc} + \alpha \sum_{t=1}^{T} H_t,$$

$$H_t = \left[ \frac{\sqrt{n}K d(x_{FP,0}^{t-1,Fair}, x_{FP,1}^{t-1,Fair})}{f(x_{FP,0}^{t-1,Fair})} + \rho_t \delta_{FP,1}^{t-1,Acc} \right],$$

$$\rho_t = \left| \frac{f(x_{FP,0}^{t-1,Fair}) - f(x_{FP,1}^{t-1,Fair})}{f(x_{FP,1}^{t-1,Fair}) f(x_{FP,0}^{t-1,Fair})} \right|.$$

### 5.2. Fair adversarial training

Theorem 5.3 provides robustness guarantee in terms of changes in predicted soft labels under the fairness attack regarding 'Opposite Direction' samples. Based on such discussion, we now derive the fairness robustness guarantee regarding fairness measures, namely DP and EOd. Consider $DP^{Fair}$, DP, $EOd^{Fair}$ and EOd as the fairness measures after and before the fairness attack, the following theorem states the fairness guarantee by static fairness and accuracy robustness under the fairness attack:

**Theorem 5.5.** *Given a classifier $f$, consider $\epsilon$-level fairness attack with step size $\alpha$ and up to $T$ iterations, let $\Delta_{sub,a}^{obj} := \max_{\{x \in \mathbb{S}_{sub,a}\}} \delta_{sub,a}^{obj}$ be the maximum shift in predicted soft labels within the subgroup under attack type obj, let $M = \max_{\{sub,a\}} M_{sub,a}$, the resulting fairness measures are upper-bounded by the corresponding clean measures and accuracy robustness of the classifier up to a multiplicative constant:*

$$DP^{Fair} \leq DP + M \cdot R_{acc}, \tag{4}$$

$$EOd^{Fair} \leq EOd + M \cdot R_{acc}, \tag{5}$$

*where $R_{acc} = (\Delta_{TP,0}^{Acc} + \min_{x_j \in \mathbb{S}_{FP,1}} (D_j^{Acc} + H_j) + \Delta_{TN,1}^{Acc} + \min_{x_j \in \mathbb{S}_{FN,0}} (D_j^{Acc} + G_j)).$*

We defer full proof to the Appendix. The DP and EOd terms in the upper-bounds of Theorem 5.5 corresponds to the static fairness, and the remainder corresponds to the accuracy robustness as stated in Theorem 5.3. Consequently, retaining small fairness violations under the fairness attack calls for two different interventions: 1) enhancing static fairness during training, which results in lower values of DP and EOd in (4) and (5); and 2) enhancing accuracy robustness, resulting in lower value of $R_{acc}$ in (4) and (5). These two interventions lead to smaller upper-bounds for $DI^{Fair}$ and $EOd^{Fair}$, thereby ensuring fairness robustness.

One direct result regarding Theorem 5.5 is to incorporate accuracy adversarial samples during training to obtain a classifier that is also robust to fairness attack. Correspondingly, regarding the defense framework against the fairness

attack, we consider the following objective to minimize the fairness gap while ensuring accuracy robustness, as means to ensure fairness robustness. Specifically, we propose to improve fairness robustness of classifier by incorporating accuracy adversarial samples and fairness constraints during training:

$$\arg\min_{\theta} \frac{1}{N} \sum_{i=1}^{N} L_{\text{CE}}(f(x_i^{\text{Acc}}), y_i), \ s.t. \ L \leq \gamma, \quad (6)$$

where $x_i^{\text{Acc}}$ corresponds to the accuracy adversarial sample by $x_i$, the $L_{\text{CE}}(f(x_i^{\text{Acc}}), y_i)$ term corresponds to accuracy robustness, and $L \leq \gamma$ corresponds to static fairness, as we derived in Theorem 5.5. $L$ can be explicitly specified by fairness relaxations during training or implicitly specified as preprocessing or post-processing techniques. We defer the pseudo-code of our fair adversarial training framework to the Appendix.

### 5.3. From fairness robustness to accuracy robustness

For the other direction, under Assumption 5.1, we have the following guarantee for accuracy robustness by fairness robustness. We will focus on $x_{\text{TP},1}$ and $x_{\text{TN},0}$, as accuracy attack regarding $x_{\text{FN},1}$ and $x_{\text{FP},0}$ does not affect accuracy, i.e., the predicted labels remain false before and after the accuracy attack.

**Theorem 5.6.** *Given a classifier $f$, consider $\epsilon$-level accuracy attack with step size $\alpha$ and up to $T$ iterations, the accuracy robustness of $x_{\text{TP},1}$ is upper-bounded by the fairness robustness of $x_{\text{TP},0}$ up to an additive constant:*

$$\delta_{TP,1}^{Acc} \leq \min_{x_{TP,0} \in \mathbb{S}_{10}} \delta_{TP,0}^{Fair} + \sum_{t=1}^{T} \sqrt{n}\alpha K d(x_{TP,1}^{t-1,Acc}, x_{TP,0}^{t-1,Acc}).$$

Here the fairness attack and the accuracy attacks are in alignment regarding $x_{\text{TP},0}$, which we use to upper-bound accuracy robustness of $x_{\text{TP},1}$. Specifically, the first term in the RHS of the inequality corresponds to the fairness robustness of $x_{\text{TP},0}$, and the second term is determined by the spatial distance between $x_{\text{TP},0}$ and $x_{\text{TP},1}$ under the accuracy attack. Theorem 5.6 shows that adversarial training w.r.t. fairness also benefits accuracy robustness. Specifically, for $f''$ under adversarial training w.r.t. fairness and $f$ under normal training, we have similar inequality, except that we now have $\delta_{\text{TP},0}^{''\text{Acc}} \leq \delta_{\text{TP},0}^{\text{Acc}}$, which indicates better accuracy robustness for $x_{\text{TP},1}$ under adversarial training. Similar upper bound also holds for TN samples:

*Remark 5.7.* For $x_{\text{TN},1}$ and $x_{\text{TN},0}$, we have similar inequality regarding the upper-bound of accuracy robustness:

$$\delta_{TN,0}^{Acc} \leq \min_{x_{TN,1} \in \mathbb{S}_{01}} \delta_{TN,1}^{Fair} + \sum_{t=1}^{T} \sqrt{n}\alpha K d(x_{TN,0}^{t-1,Acc}, x_{TN,1}^{t-1,Acc}).$$

Since the change of predictions under accuracy attack is upper-bounded by fairness robustness, it is also feasible to improve accuracy robustness of classifier by using fairness adversarial samples during training.

## 6. Experiments

We evaluate our method on four datasets: Adult (Dua & Graff, 2017), COMPAS (Larson et al., 2016), German (Dua & Graff, 2017) and CelebA (Liu et al., 2015) [1]. We use accuracy as the performance metric, and DP and EOd as fairness metrics. All the evaluation metrics are calculated based on $0-1$ cutoff, rather than the relaxed version. The classifier is chosen as ResNet-18 for CelebA and MLP for the other three datasets, and all methods are trained under the same data partition. During adversarial training, the perturbation level is set as 0.2 for Adult dataset, 0.005 for COMPAS dataset, 0.01 for German dataset and 0.1 for CelebA dataset, where the perturbation level is empirically determined to achieve the largest perturbation while still ensuring convergence.

In the following, we validate the adversarial training framework under fairness attacks and accuracy attacks, respectively. We defer details of each dataset, full experimental results and empirical verification of theoretical results to the Appendix.

**Robustness against fairness attack**. Regarding normal training, we consider the following four baselines: *Baseline* (without fairness regularization); *Preprocessing* (Yu et al., 2024); *In-processing* (Wang et al., 2022); *Post-processing* (Jang et al., 2022). Regarding adversarial training, we consider one baseline and three different realizations of fair adversarial training: *Adversarial training*: Neural network under adversarial training w.r.t. accuracy; *Adversarial training (preprocessing)* with samples reweighed by Yu et al. (2024); *Adversarial training (in-processing)* with relaxed EOd constraint by Wang et al. (2022); *Adversarial training (post-processing)* with soft predictions postprocessed by Jang et al. (2022). The three versions differ in the fairness regularization $L$ in (6). Results on classifiers under fairness attack on Adult dataset are shown in Fig. 2. The fairness attack enforces biased predictions on testing samples, and under a successful attack (DP reaches its maximum), EOd also reaches its maximum, while the accuracy is determined by the base rate of each group. Compared with adversarial training, normal training shows a sharp increase in DP and EOd under fairness attacks, and improvement in fairness under normal training does not help with fairness robustness. In comparison, classifiers under adversarial training w.r.t. accuracy show improvement in fairness robustness,

---

[1]Code available at https://github.com/cjy24/fair-adversarial-training.

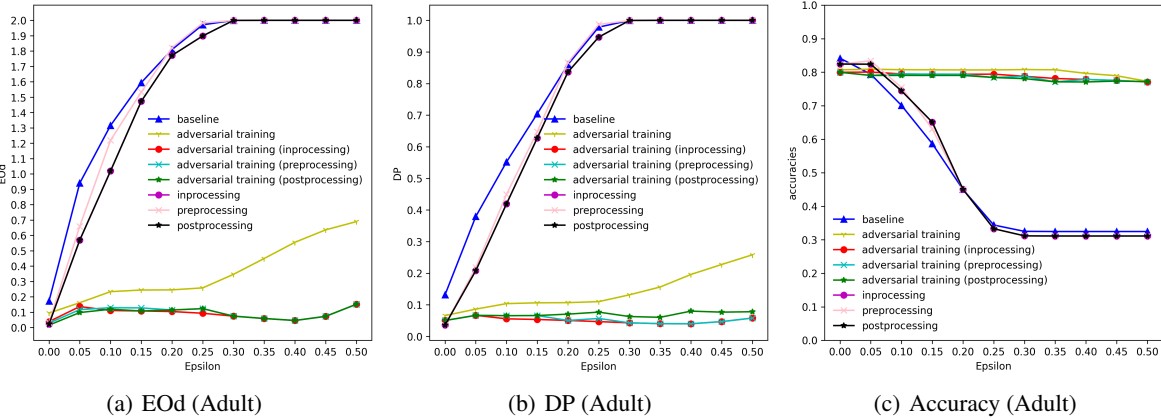

*Figure 2.* Change of accuracy, DP and EOd under fairness attack on Adult dataset. Three variations of our fair adversarial training method obtain improved fairness robustness (lower EOd, DI and higher accuracy) with significant margin.

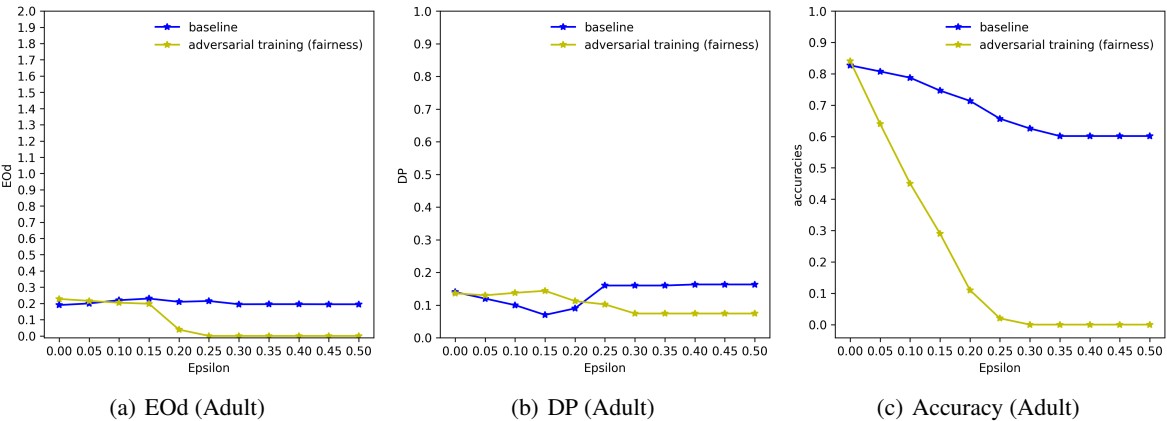

*Figure 3.* Change of accuracy, DP and EOd under accuracy attacks on Adult dataset. Adversarial training (fairness) corresponds to vanilla adversarial training using fairness adversarial samples.

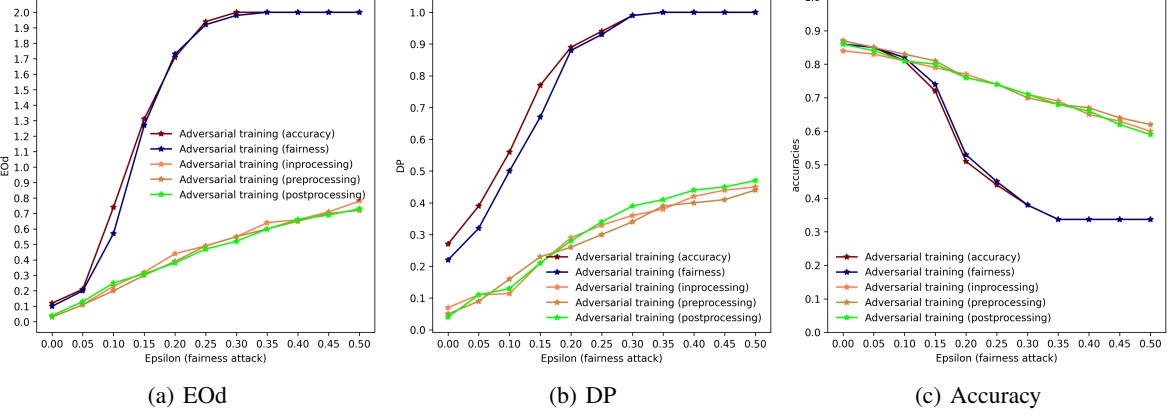

*Figure 4.* Ablation study: change of accuracy, DP and EOd on CelebA dataset under fairness attacks. Adversarial training (accuracy) corresponds to vanilla adversarial training using accuracy adversarial samples.

and classifiers under fair adversarial training show further remarkable improvement in terms of fairness robustness.

**Robustness against accuracy attack**. We move on to discuss the improvement of accuracy robustness under adversarial training w.r.t. fairness. We compare *baseline* with *adversarial training (fairness)* (adversarial training w.r.t. relaxed DP), and results on classifiers under accuracy attack on Adult dataset are shown in Fig. 3. Under a successful accuracy attack (the accuracy reaches its minimum), EOd also becomes zero, while DP does not necessarily vanish due to disparities in base rates. The classifier under adversarial training shows remarkable improvement in accuracy robustness, which validates that accuracy robustness also benefits from adversarial training w.r.t. fairness.

**Ablation study**. We validate the superiority of our method over vanilla adversarial training under fairness attacks, and the results are shown in Fig. 4. Heuristic adaptations of adversarial training utilizing fairness adversarial samples or accuracy adversarial samples alone, without incorporating fairness regularization, fail to maintain fairness robustness under larger perturbations, highlighting the intricacy of achieving fairness robustness. In comparison, our framework shows remarkable improvement in fairness robustness.

## 7. Conclusion

Fairness attack and defense are an important, yet not properly addressed problem. In this paper, we propose a unified framework for fairness attack against group fairness notions, where we show theoretically the connection of fairness attacks under different notions, and we demonstrate the connections between fairness attack and accuracy attack. We show theoretically the alignment between fairness robustness and accuracy robustness, and we propose a fair adversarial training structure, where the goal is to improve fairness robustness while maintaining fairness. Further, from experiments we validate that our method achieves better fairness robustness, and that fairness robustness and accuracy robustness align with each other.

## Acknowledgements

This work was partially supported by the EMBRIO Institute, contract #2120200, a National Science Foundation (NSF) Biology Integration Institute, and NSF IIS #1955890, IIS #2146091, IIS #2345235.

## Impact Statement

his paper aims to bridge two perspectives on adversarial robustness: robustness concerning fairness notions and robustness concerning accuracy. We believe this work will benefit the trustworthy machine learning community, particularly in contexts where both fairness and robustness are critical.

A potential consequence of our method is the generation of fairness adversarial samples, which, if not properly supervised, could be maliciously exploited to compromise trustworthiness. This potential misuse calls for proactive defenses and rigorous oversight to prevent adversarial manipulations from undermining fairness-aware AI systems.

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

# A. Experimental supplementary

We validate our method on four datasets:

- Adult (Dua & Graff, 2017): The Adult dataset contains 65,123 samples with 14 attributes. The goal is to predict whether an individual's annual income exceeds $50K$, and the sensitive attribute is chosen as *race*.

- COMPAS (Larson et al., 2016): The ProPublica COMPAS dataset contains 7,215 samples with 10 attributes. The goal is to predict whether a defendant re-offend within two years. Following the protocol in earlier fairness methods (Zafar et al., 2017), we only select white and black individuals in COMPAS dataset, which contains 6,150 samples in total. The sensitive attribute in this dataset is *race*.

- German (Dua & Graff, 2017): The German credit risk dateset contains 1,000 samples with 9 attributes. The goal is to predict whether a client is highly risky, and the sensitive attribute in this dataset is *sex*.

- CelebA (Liu et al., 2015): CelebA dataset contains 202,599 samples with 40 binary attributes. We choose gender as target label, and the sensitive attribute in this dataset is *age*.

Details of our fair adversarial training framework are shown as follows:

- *Adversarial training (preprocessing (Yu et al., 2024))*: Neural network under adversarial training w.r.t. accuracy with training samples reweighed by Yu et al. (2024);

- *Adversarial training (in-processing (Wang et al., 2022))*: Neural network under adversarial training w.r.t. accuracy with relaxed EOd constraint by Wang et al. (2022);

- *Adversarial training (post-processing (Jang et al., 2022))*: Neural network under adversarial training w.r.t. accuracy with predicted soft labels postprocessed by Jang et al. (2022).

# B. Pseudo-code

We include pseudo-codes of the three variations of our fair adversarial training framework as follows:

---
**Algorithm 1** Adversarial training (preprocessing)

---
**Input:** Network $f_\theta$ with parameter $\theta$, training data $D = \{(\hat{x}_i, \hat{y}_i), 1 \leqslant i \leqslant N\}$ reweighed inversely proportional to each group–label pair's empirical frequency by (Yu et al., 2024), batch size $M$, training epochs $E$, learning rate $\eta$, allowed perturbation set $S = \{\epsilon, \|\epsilon\| \leqslant \epsilon_0\}$, attack step size $\alpha$, attack steps $k$.

1: **for** epoch $= 1$ **to** $E$ **do**:
2:     **for** batch $B = \{(\hat{x}_j, \hat{y}_j), 1 \leqslant j \leqslant M\} \subset D$ **do**:
3:         **for** attack step $t = 0$ **to** $k - 1$ **do**:
4:

$$\hat{x}_j^{t+1} = \Pi_{\hat{x}_j + S}\left(\hat{x}_j^t + \alpha \operatorname{sign}\left(\nabla_{\hat{x}_j} L_{\text{CE}}(\hat{x}_j^t, \hat{y}_j)\right)\right).$$

5:         **end for**
6:         Training loss:

$$L_{\text{CE}} = \frac{1}{M} \sum_{j=1}^{M} -\hat{y}_j \log(f_\theta(\hat{x}_j^k)) - (1 - \hat{y}_j) \log(1 - f_\theta(\hat{x}_j^k)).$$

7:         Update $\theta$ by gradient descent:

$$\theta = \theta - \eta \frac{L_{\text{CE}}}{\partial \theta}.$$

8:     **end for**
9: **end for**

---

---

**Algorithm 2** Adversarial training (in-processing)

---

**Input:** Network $f_\theta$ with parameter $\theta$, training data $D = \{(x_i, y_i, a_i), 1 \leqslant i \leqslant N\}$, batch size $M$, training epochs $E$, learning rate $\eta$, allowed perturbation set $S = \{\epsilon, \|\epsilon\| \leqslant \epsilon_0\}$, attack step size $\alpha$, attack steps $k$, regularization coefficient $\lambda$, gorup size $|\mathbb{S}_{.0}|, |\mathbb{S}_{.1}|$.

---

1: **for** epoch $= 1$ **to** $E$ **do**:
2:     **for** batch $B = \{(x_j, y_j), 1 \leqslant i \leqslant M\} \subset D$ **do**:
3:         **for** attack step $t = 0$ **to** $k - 1$ **do**:
4:

$$x_j^{t+1} = \Pi_{x_j+S}\left(x_j^t + \alpha \operatorname{sign}\left(\nabla_{x_j} L_{\text{CE}}(x_j^t, y_j)\right)\right).$$

5:         **end for**
6:     Training loss:

$$L_{\text{CE}} = \frac{1}{M} \sum_{j=1}^{M} -y_j \log(f_\theta(x_j^k)) - (1 - y_j) \log(1 - f_\theta(x_j^k)),$$

$$L_{\text{DP}} = \left| \sum_{x_j \in \mathbb{S}_{.1}} \frac{f(x_j)}{|\mathbb{S}_{.1}|} - \sum_{x_j \in \mathbb{S}_{.0}} \frac{f(x_j)}{|\mathbb{S}_{.0}|} \right|,$$

$$L_{\text{total}} = L_{\text{CE}} + \lambda L_{\text{DP}}.$$

7:     Update $\theta$ by gradient descent:

$$\theta = \theta - \eta \frac{L_{\text{total}}}{\partial \theta}.$$

8:     **end for**
9: **end for**

---

---

**Algorithm 3** Adversarial training (post-processing)

---

**Input:** Network $f_\theta$ with parameter $\theta$, training data $D = \{(x_i, y_i, a_i), 1 \leqslant i \leqslant N\}$, batch size $M$, training epochs $E$, learning rate $\eta$, allowed perturbation set $S = \{\epsilon, \|\epsilon\| \leqslant \epsilon_0\}$, attack step size $\alpha$, attack steps $k$.

---

1: **for** epoch $= 1$ **to** $E$ **do**:
2:     **for** batch $B = \{(x_j, y_j), 1 \leqslant i \leqslant M\} \subset D$ **do**:
3:         **for** attack step $t = 0$ **to** $k - 1$ **do**:
4:

$$x_j^{t+1} = \Pi_{x_j+S}\left(x_j^t + \alpha \operatorname{sign}\left(\nabla_{x_j} L_{\text{CE}}(x_j^t, y_j)\right)\right).$$

5:         **end for**
6:     Training loss:

$$L_{\text{CE}} = \frac{1}{M} \sum_{j=1}^{M} -y_j \log(f_\theta(x_j^k)) - (1 - y_j) \log(1 - f_\theta(x_j^k)).$$

7:     Update $\theta$ by gradient descent:

$$\theta = \theta - \eta \frac{L_{\text{CE}}}{\partial \theta}.$$

8:     **end for**
9: **end for**
10: post-process $f_\theta(x)$ on training data $D$ by estimating the group-dependent thresholds (Jang et al., 2022).

---

## C. Connection between DP attack and EOd attack

We empirically validate our discussion regarding the relationship between DP and EOd attack as in Corollary 4.1. As shown in Fig 2, under a successful DP attack, EOd always reaches its maximum, and a successful DP attack also leads to a

successful EOd attack.

## D. Proof of Corollary 4.1

*Proof.* The objective for EOd attack can be written as the following form:

$$
\begin{aligned}
L_{\text{EOd}} &= \left| \sum_{x_i \in \mathbb{S}_{00}} \frac{f(x_i)}{|\mathbb{S}_{00}|} - \sum_{x_i \in \mathbb{S}_{01}} \frac{f(x_i)}{|\mathbb{S}_{01}|} \right| + \left| \sum_{x_i \in \mathbb{S}_{10}} \frac{f(x_i)}{|\mathbb{S}_{10}|} - \sum_{x_i \in \mathbb{S}_{11}} \frac{f(x_i)}{|\mathbb{S}_{11}|} \right| \\
&\geq \left| \sum_{x \in \mathbb{S}_{00}} \frac{f(x)}{|\mathbb{S}_{00}|} - \sum_{x \in \mathbb{S}_{01}} \frac{f(x)}{|\mathbb{S}_{01}|} + \sum_{x \in \mathbb{S}_{10}} \frac{f(x)}{|\mathbb{S}_{10}|} - \sum_{x \in \mathbb{S}_{11}} \frac{f(x)}{|\mathbb{S}_{11}|} \right| \\
&= \left| \sum_{x \in \mathbb{S}_{00}} \frac{|\mathbb{S}_{.0}|}{|\mathbb{S}_{00}|} \frac{f(x)}{|\mathbb{S}_{.0}|} + \sum_{x \in \mathbb{S}_{10}} \frac{|\mathbb{S}_{.0}|}{|\mathbb{S}_{10}|} \frac{f(x)}{|\mathbb{S}_{.0}|} - \sum_{x \in \mathbb{S}_{01}} \frac{|\mathbb{S}_{.1}|}{|\mathbb{S}_{01}|} \frac{f(x)}{|\mathbb{S}_{.1}|} - \sum_{x \in \mathbb{S}_{11}} \frac{|\mathbb{S}_{.1}|}{|\mathbb{S}_{11}|} \frac{f(x)}{|\mathbb{S}_{.1}|} \right|.
\end{aligned}
$$

This shows that the EOd attack is lower-bounded by the weighted DP attack as in (1). Specifically, under a successful DP attack, we have $f(x) \geq 0.5, \forall x \in \mathbb{S}_{.a}$ and $f(x) \leq 0.5, \forall x \in \mathbb{S}_{.a'}$, and we have EOd under under such attack as

$$
\text{EOd} = \left| \frac{\sum_{x \in \mathbb{S}_{10}} \mathbb{1}[f(x) \geq 0.5]}{|\mathbb{S}_{10}|} - \frac{\sum_{x \in \mathbb{S}_{11}} \mathbb{1}[f(x) \geq 0.5]}{|\mathbb{S}_{11}|} \right| + \left| \frac{\sum_{x \in \mathbb{S}_{00}} \mathbb{1}[f(x) < 0.5]}{|\mathbb{S}_{00}|} - \frac{\sum_{x \in \mathbb{S}_{01}} \mathbb{1}[f(x) < 0.5]}{|\mathbb{S}_{01}|} \right| = 2,
\tag{7}
$$

which shows that a successful DP attack always implies a successful EOd attack.

*Remark* D.1. A successful EOd attack does not always imply a successful DP attack. Consider the following counter-example: assume $\sum_{x_i \in \mathbb{S}_{00}} \frac{f(x_i)}{|\mathbb{S}_{00}|} \leq \sum_{x_i \in \mathbb{S}_{01}} \frac{f(x_i)}{|\mathbb{S}_{01}|}$ and $\sum_{x_i \in \mathbb{S}_{10}} \frac{f(x_i)}{|\mathbb{S}_{10}|} \geq \sum_{x_i \in \mathbb{S}_{11}} \frac{f(x_i)}{|\mathbb{S}_{11}|}$, under a successful EOd attack, all the predictions in the disadvantaged group will become correct, while all the predictions in the advantaged group will become incorrect, and the disparate impact will not be maximized as both groups contain positive predictions.

$\square$

## E. Proof of Corollary 4.2

*Proof.* The objective for accuracy attack for sample $x_i$ can be written as

$$
\max_{\delta} L_{\text{CE}}((x_i + \epsilon), y_i), \|\epsilon\| \leq \epsilon_0,
\tag{8}
$$

Consider the DP attack in (1), we have the objective for DP attack as follows:

$$
\max_{\delta} \alpha_i \frac{f(x_i + \epsilon)}{|\mathbb{S}_{.a_i}|}, \|\epsilon\| \leq \epsilon',
$$

where $\alpha_i = -1$ for $a_i = 0$ and $\alpha_i = 1$ for $a_i = 1$. For positive samples, we can further write (8) as

$$
\max_{\delta} - \log(f(x_i + \epsilon)), \|\epsilon\| \leq \epsilon_0,
$$

where the perturbation is expected to minimize the predicted soft label, which is in alignment with the objective of DP when $\alpha_i = -1$, i.e., for TP and FN disadvantaged samples, the two attacks are in alignment. Similarly, for negative samples, we have (8) as

$$
\max_{\delta} - \log(1 - f(x + \epsilon)), \|\epsilon\| \leq \epsilon',
$$

where the perturbation is expected to maximize the predicted soft label, which is in alignment with the objective of DP when $\alpha_i = 1$, i.e., for TN and FP advantaged samples, the two attacks are in alignment. Specifically, for gradient-based attacks, we have the two kinds of attack equivalent. $\square$

# F. Proof of Theorem 5.3

*Proof.* Let $f$ be the function of classifier, consider the positive testing set $\{(x_i, 1, a_i), 1 \leq i \leq N\}$ for simplicity, at $t$-th iteration, we have the linear approximation of testing CE loss under the fairness attack as follows:

$$L_{\text{CE}}(x^t) = -\log(f(x^t)) = -\log(f(x^{t-1}) - \delta^{t-1,\text{Fair}}) = -\log(f(x^{t-1})) + \frac{\delta^{t-1,\text{Fair}}}{f(x^{t-1})} + r_L(x^{t-1}), \tag{9}$$

where $\delta^{t-1,\text{Fair}}$ is the change of soft label induced by the fairness attack at $t$-th iteration, and $r_L(x)$ is the remainder of Taylor's expansion. For gradient-based attack, the predicted soft label for fairness adversarial sample can be formulated as

$$f(x^t) = f(x^{t-1} + \alpha\text{sign}(\nabla_{x^{t-1}} L_{\text{DP}})) = f(x^{t-1}) + \alpha(\nabla_{x^{t-1}} f(x^{t-1}))^T \text{sign}(\nabla_{x^{t-1}} L_{\text{DP}}) + r_f(x^{t-1}), \tag{10}$$

where $L_{\text{DP}}$ is the relaxed DP and $r_f(x)$ is the remainder of Taylor's expansion. Let $D^{t,\text{Fair}} := |L(x^t) - L(x^{t-1})|$ be the change of CE loss under the fairness attack at $t$-th iteration, according to (9) and (10) we have

$$\begin{aligned}
D^{t,\text{Fair}} &= \left| L_{\text{CE}}(x^t) - L_{\text{CE}}(x^{t-1}) \right| = \left| -\log(f(x^{t-1})) + \frac{\delta^{t-1,\text{Fair}}}{f(x^{t-1})} + r_L(x) + \log(f(x)) \right| \\
&\approx \frac{|\alpha(\nabla_{x^{t-1}} f(x^{t-1}))^T \text{sign}(\nabla_{x^{t-1}} L_{\text{DP}})|}{f(x^{t-1})}.
\end{aligned}$$

Consider FN sample $x_{\text{FN},0}$ from disadvantaged group and FN sample $x_{\text{FN},1}$ from advantaged group, since the gradient of $f$ w.r.t. $x$ is Lipschitz with constant $K$, we have the difference of change in CE loss under DP attack at $t$-th iteration as follows:

$$\begin{aligned}
&|D_{\text{FN},1}^{t,\text{Fair}} - D_{\text{FN},0}^{t,\text{Fair}}| \\
=& \alpha \left| \frac{|(\nabla_{x_{\text{FN},1}^{t-1,\text{Fair}}} f(x_{\text{FN},1}^{t-1,\text{Fair}}))^T \text{sign}(\nabla_{x_{\text{FN},1}^{t-1,\text{Fair}}} L_{\text{DP}})|}{f(x_{\text{FN},1}^{t-1,\text{Fair}})} - \frac{|(\nabla_{x_{\text{FN},0}^{t-1,\text{Fair}}} f(x_{\text{FN},0}^{t-1,\text{Fair}}))^T \text{sign}(\nabla_{x_{\text{FN},0}^{t-1,\text{Fair}}} L_{\text{DP}})|}{f(x_{\text{FN},0}^{t-1,\text{Fair}})} \right| \\
=& \alpha \left| \frac{(\nabla_{x_{\text{FN},1}^{t-1,\text{Fair}}} f(x_{\text{FN},1}^{t-1,\text{Fair}}))^T \text{sign}(\nabla_{x_{\text{FN},1}^{t-1,\text{Fair}}} L_{\text{DP}})}{f(x_{\text{FN},1}^{t-1,\text{Fair}})} + \frac{(\nabla_{x_{\text{FN},0}^{t-1,\text{Fair}}} f_\theta(x_{\text{FN},0}^{t-1,\text{Fair}}))^T \text{sign}(\nabla_{x_{\text{FN},0}^{t-1,\text{Fair}}} L_{\text{DP}})}{f(x_{\text{FN},0}^{t-1,\text{Fair}})} \right| \\
=& \alpha \left| \frac{(\nabla_{x_{\text{FN},1}^{t-1,\text{Fair}}} f(x_{\text{FN},1}^{t-1,\text{Fair}}))^T \text{sign}(\frac{1}{N_1} \nabla_{x_{\text{FN},1}^{t-1,\text{Fair}}} f(x_{\text{FN},1}^{t-1,\text{Fair}}))}{f_\theta(x_{\text{FN},1}^{t-1,\text{Fair}})} - \frac{(\nabla_{x_{\text{FN},0}^{t-1,\text{Fair}}} f(x_{\text{FN},0}^{t-1,\text{Fair}}))^T \text{sign}(\frac{1}{N_0} \nabla_{x_{\text{FN},0}^{t-1,\text{Fair}}} f(x_{\text{FN},0}^{t-1,\text{Fair}}))}{f(x_{\text{FN},0}^{t-1,\text{Fair}})} \right| \\
=& \alpha \left| \frac{\sum_{j=1}^n |\partial_{x_j} f(x_{\text{FN},1}^{t-1,\text{Fair}})|}{f(x_{\text{FN},1}^{t-1,\text{Fair}})} - \frac{\sum_{j=1}^n |\partial_{x_j} f(x_{\text{FN},0}^{t-1,\text{Fair}})|}{f(x_{\text{FN},0}^{t-1,\text{Fair}})} \right| \\
=& \alpha \left| \frac{\|\nabla_{x_{\text{FN},1}^{t-1,\text{Fair}}} f(x_{\text{FN},1}^{t-1,\text{Fair}})\|_1}{f(x_{\text{FN},1}^{t-1,\text{Fair}})} - \frac{\|\nabla_{x_{\text{FN},0}^{t-1,\text{Fair}}} f(x_{\text{FN},0}^{t-1,\text{Fair}})\|_1}{f(x_{\text{FN},0}^{t-1,\text{Fair}})} \right|,
\end{aligned} \tag{11}$$

where $n$ is the dimension of input feature. Since $\nabla_x f(x)$ is Lipschitz, we have

$$\|\nabla_x f(x_1)\|_2 - \|\nabla_x f(x_0)\|_2 \leq \|\nabla_x f(x_1) - \nabla_x f(x_0)\|_2 \leq Kd(x_1, x_0),$$

where the first sign is due to triangle inequality. By Jensen's inequality we have $\|x\|_2 \leq \|x\|_1 \leq \sqrt{n}\|x\|_2$, and

$$\|\nabla_x f(x_1)\|_1 - \|\nabla_x f(x_0)\|_1 \leq \|\nabla_x f(x_0) - \nabla_x f(x_1)\|_1 \leq \sqrt{n}Kd(x_1, x_0). \tag{12}$$

Assume $\dfrac{\|\nabla_{x_{\text{FN},1}^{t-1,\text{Fair}}} f(x_{\text{FN},1}^{t-1,\text{Fair}})\|_1}{f(x_{\text{FN},1}^{t-1,\text{Fair}})} \geq \dfrac{\|\nabla_{x_{\text{FN},0}^{t-1,\text{Fair}}} f(x_{\text{FN},0}^{t-1,\text{Fair}})\|_1}{f(x_{\text{FN},0}^{t-1,\text{Fair}})}$, plugging (12) back into (11), we have

$$
\begin{aligned}
&|D_{\text{FN},1}^{t,\text{Fair}} - D_{\text{FN},0}^{t,\text{Fair}}| \\
=&\alpha \left| \frac{\|\nabla_{x_{\text{FN},1}^{t-1,\text{Fair}}} f(x_{\text{FN},1}^{t-1,\text{Fair}})\|_1}{f(x_{\text{FN},1}^{t-1,\text{Fair}})} - \frac{\|\nabla_{x_{\text{FN},0}^{t-1,\text{Fair}}} f(x_{\text{FN},0}^{t-1,\text{Fair}})\|_1}{f(x_{\text{FN},0}^{t-1,\text{Fair}})} \right| \\
\leq&\alpha \left| \frac{\sqrt{n}Kd(x_{\text{FN},1}^{t-1,\text{Fair}}, x_{\text{FN},0}^{t-1,\text{Fair}}) + \|\nabla_{x_{\text{FN},0}^{t-1,\text{Fair}}} f(x_{\text{FN},0}^{t-1,\text{Fair}})\|_1}{f(x_{\text{FN},1}^{t-1,\text{Fair}})} - \frac{\|\nabla_{x_{\text{FN},0}^{t-1,\text{Fair}}} f(x_{\text{FN},0}^{t-1,\text{Fair}})\|_1}{f(x_{\text{FN},0}^{t-1,\text{Fair}})} \right| \\
\leq& \frac{\sqrt{n}\alpha Kd(x_{\text{FN},1}^{t-1,\text{Fair}}, x_{\text{FN},0}^{t-1,\text{Fair}})}{f(x_{\text{FN},1}^{t-1,\text{Fair}})} + \left| \frac{\alpha\|\nabla_{x_{\text{FN},0}^{t-1,\text{Fair}}} f(x_{\text{FN},0}^{t-1,\text{Fair}})\|_1}{f(x_{\text{FN},1}^{t-1,\text{Fair}})} - \frac{\alpha\|\nabla_{x_{\text{FN},0}^{t-1,\text{Fair}}} f(x_{\text{FN},0}^{t-1,\text{Fair}})\|_1}{f(x_{\text{FN},0}^{t-1,\text{Fair}})} \right|,
\end{aligned}
\tag{13}
$$

where $d(x,y) \coloneqq \|x-y\|_2$ is the distance between the two feature. Taking the summation over $T$ iterations, we have

$$
|D_{\text{FN},1}^{\text{Fair}} - D_{\text{FN},0}^{\text{Fair}}| \leq \sum_{t=1}^{T} \frac{\sqrt{n}\alpha Kd(x_{\text{FN},1}^{t-1,\text{Fair}}, x_{\text{FN},0}^{t-1,\text{Fair}})}{f(x_{\text{FN},1}^{t-1,\text{Fair}})} + \sum_{t=1}^{T} \alpha \left| \frac{f(x_{\text{FN},0}^{t-1,\text{Fair}}) - f(x_{\text{FN},1}^{t-1,\text{Fair}})}{f(x_{\text{FN},1}^{t-1,\text{Fair}})f(x_{\text{FN},0}^{t-1,\text{Fair}})} \right| \delta_{\text{FN},0}^{t-1,\text{Acc}},
\tag{14}
$$

where $\delta_{\text{FN},0}^{t-1,\text{Acc}} \coloneqq \|\nabla_{x_{\text{FN},0}^{t-1,\text{Fair}}} f_\theta(x_{\text{FN},0}^{t-1,\text{Fair}})\|_1$ is the change of $x_{\text{FN},0}$'s predicted label under $\epsilon$-level accuracy attack at $t$-th iteration since both are equivalent regarding $x_{\text{FN},0}$. Since the above inequality holds true for all disadvantaged TP samples and $D_{\text{FN},1}^{\text{Acc}} = D_{\text{FN},1}^{\text{Fair}}$, we can further write (14) as

$$
D_{\text{FN},1}^{\text{Fair}} \leq \min_{x_{\text{FN},0} \in \mathbb{S}_{10}} D_{\text{FN},0}^{\text{Acc}} + \sum_{t=1}^{T} \frac{\sqrt{n}\alpha Kd(x_{\text{FN},1}^{t-1,\text{Fair}}, x_{\text{FN},0}^{t-1,\text{Fair}})}{f(x_{\text{FN},1}^{t-1,\text{Fair}})} + \sum_{t=1}^{T} \alpha \left| \frac{f(x_{\text{FN},0}^{t-1,\text{Fair}}) - f(x_{\text{FN},1}^{t-1,\text{Fair}})}{f(x_{\text{FN},1}^{t-1,\text{Fair}})f(x_{\text{FN},0}^{t-1,\text{Fair}})} \right| \delta_{\text{FN},0}^{t-1,\text{Acc}}.
$$

This shows that under the fairness attack, the difference of change in performance regarding marginal advantaged FN samples are upper-bounded by the robustness of marginal disadvantaged FN samples up to an additive constant. For $f$ under normal training and $f'$ under normal training, we have similar upper-bound except that we now have $\delta_{\text{FN},0}^{'t-1,\text{Acc}} \geq \delta_{\text{FN},0}^{t-1,\text{Acc}}$, which indicates that the adversarial classifier achieves tighter upper-bound than that of a normal classifier. For $\dfrac{\|\nabla_{x_{\text{FN},1}^{t-1,\text{Fair}}} f(x_{\text{FN},1}^{t-1,\text{Fair}})\|_1}{f(x_{\text{FN},1}^{t-1,\text{Fair}})} \leq \dfrac{\|\nabla_{x_{\text{FN},0}^{t-1,\text{Fair}}} f(x_{\text{FN},0}^{t-1,\text{Fair}})\|_1}{f(x_{\text{FN},0}^{t-1,\text{Fair}})}$, we have same upper-bound:

$$
\begin{aligned}
&|D_{\text{FN},1}^{t,\text{Fair}} - D_{\text{FN},0}^{t,\text{Fair}}| \\
=&\alpha \left| \frac{\|\nabla_{x_{\text{FN},1}^{t-1,\text{Fair}}} f(x_{\text{FN},1}^{t-1,\text{Fair}})\|_1}{f(x_{\text{FN},1}^{t-1,\text{Fair}})} - \frac{\|\nabla_{x_{\text{FN},0}^{t-1,\text{Fair}}} f(x_{\text{FN},0}^{t-1,\text{Fair}})\|_1}{f(x_{\text{FN},0}^{t-1,\text{Fair}})} \right| \\
\leq&\alpha \left| \frac{\|\nabla_{x_{\text{FN},0}^{t-1,\text{Fair}}} f(x_{\text{FN},0}^{t-1,\text{Fair}})\|_1}{f(x_{\text{FN},0}^{t-1,\text{Fair}})} - \frac{\|\nabla_{x_{\text{FN},0}^{t-1,\text{Fair}}} f(x_{\text{FN},0}^{t-1,\text{Fair}})\|_1 - \sqrt{n}Kd(x_{\text{FN},1}^{t-1,\text{Fair}}, x_{\text{FN},0}^{t-1,\text{Fair}})}{f(x_{\text{FN},0}^{t-1,\text{Fair}})} \right| \\
\leq& \frac{\sqrt{n}\alpha Kd(x_{\text{FN},1}^{t-1,\text{Fair}}, x_{\text{FN},0}^{t-1,\text{Fair}})}{f(x_{\text{FN},1}^{t-1,\text{Fair}})} + \left| \frac{\alpha\|\nabla_{x_{\text{FN},0}^{t-1,\text{Fair}}} f(x_{\text{FN},0}^{t-1,\text{Fair}})\|_1}{f(x_{\text{FN},1}^{t-1,\text{Fair}})} - \frac{\alpha\|\nabla_{x_{\text{FN},0}^{t-1,\text{Fair}}} f(x_{\text{FN},0}^{t-1,\text{Fair}})\|_1}{f(x_{\text{FN},0}^{t-1,\text{Fair}})} \right|.
\end{aligned}
$$

$\square$

## G. Proof of Theorem 5.5

*Proof.* Let $e_{ya}$ be the error rate in the subgroup $\mathbb{S}_{ya}$, let $b_a$ be the base rate in group $a$, denote as $\text{mea}^{\text{Fair}}$ the fairness measure mea after the fairness attack, we have the following expression regarding DI after the fairness attack:

$$
\begin{aligned}
\text{DP}^{\text{Fair}} &= \left| \int_{0.5}^{1} p_0^{\text{Fair}} df - \int_{0.5}^{1} p_1^{\text{Fair}} df \right| \\
&= \left| \int_{0.5}^{1} \left( b_0(1-e_{10}^{\text{Fair}}) p_{\text{TP},0}^{\text{Fair}} + (1-b_0) e_{00}^{\text{Fair}} p_{\text{FP},0}^{\text{Fair}} \right) df - \int_{0.5}^{1} \left( b_1(1-e_{11}^{\text{Fair}}) p_{\text{TP},1}^{\text{Fair}} + (1-b_1) e_{01}^{\text{Fair}} p_{\text{FP},1}^{\text{Fair}} \right) df \right| \\
&\leq \left| \int_{0.5+\Delta_{\text{TP},0}^{\text{Fair}}}^{1} b_0(1-e_{10}) p_{\text{TP},0} df + \int_{0.5+\Delta_{\text{FP},0}^{\text{Fair}}}^{1} (1-b_0) e_{00} p_{\text{FP},0} df \right. \\
&\quad - \int_{0.5}^{1} b_1(1-e_{11}) p_{\text{TP},1} df - \int_{0.5}^{1} (1-b_1) e_{01} p_{\text{FP},1} df \\
&\quad \left. - \int_{0.5-\Delta_{\text{TN},1}^{\text{Fair}}}^{0.5} (1-b_1)(1-e_{01}) p_{\text{TN},1} df - \int_{0.5-\Delta_{\text{FN},1}^{\text{Fair}}}^{0.5} b_1 e_{11} p_{\text{FN},1} df \right| \\
&= \left| b_1(1-e_{11}) P_{\text{TP},1}(0.5) + (1-b_1) e_{01} P_{\text{FP},1}(0.5) + (1-b_1)(1-e_{01}) P_{\text{TN},1}(0.5 - \Delta_{\text{TN},1}^{\text{Fair}}) \right. \\
&\quad \left. + b_1 e_{11} P_{\text{FN},1}(0.5 - \Delta_{\text{FN},1}^{\text{Fair}}) - b_0(1-e_{10}) P_{\text{TP},0}(0.5 + \Delta_{\text{TP},0}^{\text{Fair}}) - (1-b_0) e_{00} P_{\text{FP},0}(0.5 + \Delta_{\text{FP},0}^{\text{Fair}}) \right|,
\end{aligned}
\tag{15}
$$

where $\Delta_{\text{sub},a}^{\text{Fair}} := \max_{i \in \{\text{sub},a\}} \delta_i^{\text{Fair}}$ is the maximum prediction shift within the subgroup, $P_{\text{sub},a}$ is the CDF of $p_{\text{sub},a}$, and the inequality is due to that the worst-case prediction shift upper-bounds the overall shift in the distribution of predicted soft label. Since $P_{\text{sub},a}$ is Lipschitz continuous with constant $M_{\text{sub},a}$ ($p_{\text{sub},a}$ is uniformly bounded by $M_{\text{sub},a}$), we can further simplify (15) as

$$
\begin{aligned}
\text{DP}^{\text{Fair}} &\leq \left| b_1(1-e_{11}) P_{\text{TP},1}(0.5) + (1-b_1) e_{01} P_{\text{FP},1}(0.5) + (1-b_1)(1-e_{01}) P_{\text{TN},1}(0.5 - \Delta_{\text{TN},1}^{\text{Fair}}) + b_1 e_{11} P_{\text{FN},1}(0.5 - \Delta_{\text{FN},1}^{\text{Fair}}) \right. \\
&\quad \left. - b_0(1-e_{10}) P_{\text{TP},0}(0.5 + \Delta_{\text{TP},0}^{\text{Fair}}) - (1-b_0) e_{00} P_{\text{FP},0}(0.5 + \Delta_{\text{FP},0}^{\text{Fair}}) \right| \\
&\leq \text{DP} + b_0(1-e_{10}) M_{\text{TP},0} \Delta_{\text{TP},0}^{\text{Fair}} + (1-b_0) e_{00} M_{\text{FP},0} \Delta_{\text{FP},0}^{\text{Fair}} + (1-b_1)(1-e_{01}) M_{\text{TN},1} \Delta_{\text{TN},1}^{\text{Fair}} + b_1 e_{11} M_{\text{FN},1} \Delta_{\text{FN},1}^{\text{Fair}} \\
&\leq \text{DP} + M \left( \Delta_{\text{TP},0}^{\text{Acc}} + \min_{j \in \mathbb{S}_{\text{FP},1}} (D_j^{\text{Acc}} + H_j) + \Delta_{\text{TN},1}^{\text{Acc}} + \min_{j \in \mathbb{S}_{\text{FN},0}} (D_j^{\text{Acc}} + G_j) \right),
\end{aligned}
$$

where $M = \max\{M_{\text{TP},0}, M_{\text{FP},0}, M_{\text{TN},1}, M_{\text{FN},1}\}$, and the two minimization terms in the last inequality correspond to the upper-bounds in Theorem 5.3 and Remark 5.4. Since the fairness robustness and accuracy robustness are equivalent regarding $x_{\text{TP},0}$ and $_{\text{TN},1}$, and $D_j$, $H_j$ and $G_j$ are determined by the intrinsic distance between samples and the accuracy robustness of $x_{\text{FP},1}$ and $x_{\text{FN},0}$, we can conclude that $\text{DP}^{\text{Fair}}$ is upper-bounded by static fairness, i.e., the DP term, and the accuracy robustness $\delta_{\text{TP},0}^{\text{Acc}}$, $\min_{j \in \mathbb{S}_{\text{FP},1}} (D_j^{\text{Acc}} + H_j)$, $\Delta_{\text{TN},1}^{\text{Acc}}$ and $\min_{j \in \mathbb{S}_{\text{FN},0}} (D_j^{\text{Acc}} + G_j)$, which validates our fair adversarial training framework.

Similarly, we have the following upper-bound regarding $\text{EOd}^{\text{Fair}}$:

$$\text{EOd}^{\text{Fair}}$$

$$= \left| \int_{0.5}^{1} p_{00}^{\text{Fair}} df - \int_{0.5}^{1} p_{01}^{\text{Fair}} df \right| + \left| \int_{0.5}^{1} p_{10}^{\text{Fair}} df - \int_{0.5}^{1} p_{11}^{\text{Fair}} df \right|$$

$$= \left| \int_{0.5}^{1} \left( (1 - e_{00}^{\text{Fair}}) p_{\text{TN},0}^{\text{Fair}} + e_{00}^{\text{Fair}} p_{\text{FP},0}^{\text{Fair}} \right) df - \int_{0.5}^{1} \left( (1 - e_{01}') p_{\text{TN},1}^{\text{Fair}} + e_{01}^{\text{Fair}} p_{\text{FP},1}^{\text{Fair}} \right) df \right|$$

$$+ \left| \int_{0.5}^{1} \left( (1 - e_{10}^{\text{Fair}}) p_{\text{TP},0}^{\text{Fair}} + e_{10}^{\text{Fair}} p_{\text{FN},0}^{\text{Fair}} \right) df - \int_{0.5}^{1} \left( (1 - e_{11}^{\text{Fair}}) p_{\text{TP},1}^{\text{Fair}} + e_{11}^{\text{Fair}} p_{\text{FN},1}^{\text{Fair}} \right) df \right|$$

$$= \left| \int_{0.5}^{1} (1 - e_{00}) p_{\text{TN},0} df + \int_{0.5 + \Delta_{\text{FP},0}^{\text{Fair}}}^{1} e_{00} p_{\text{FP},0} df - \int_{0.5 - \Delta_{\text{TN},1}^{\text{Fair}}}^{1} (1 - e_{01}) p_{\text{TN},1} df - \int_{0.5}^{1} e_{01} p_{\text{FP},1} df \right|$$

$$+ \left| \int_{0.5 + \Delta_{\text{TP},0}^{\text{Fair}}}^{1} (1 - e_{10}) p_{\text{TP},0} df + \int_{0.5}^{1} e_{10} p_{\text{FN},0} df - \int_{0.5}^{1} (1 - e_{11}) p_{\text{TP},1} df - \int_{0.5 - \Delta_{\text{FN},1}^{\text{DI}}}^{1} e_{11} p_{\text{FN},1} df \right|$$

$$\leq \text{EOd} + e_{00} M_{\text{FP},0} \Delta_{\text{FP},0}^{\text{Fair}} + (1 - e_{01}) M_{\text{TN},1} \Delta_{\text{TN},1}^{\text{Fair}} + (1 - e_{10}) M_{\text{TP},0} \Delta_{\text{TP},0}^{\text{Fair}} + e_{11} M_{\text{FN},1} \Delta_{\text{FN},1}^{\text{Fair}}$$

$$\leq \text{EOd} + M((\Delta_{\text{TP},0}^{\text{Acc}} + \min_{j \in \mathbb{S}_{\text{FP},1}} (D_j^{\text{Acc}} + H_j) + \Delta_{\text{TN},1}^{\text{Acc}} + \min_{j \in \mathbb{S}_{\text{FN},0}} (D_j^{\text{Acc}} + G_j))),$$

where the first term in the last inequality corresponds to static fairness, i.e., EOd without fairness perturbation, and the second term corresponds to accuracy robustness. $\qquad\square$

## H. Proof of Theorem 5.6

*Proof.* Let $f$ be the function of classifier, consider $x_{\text{TP},0}$, we have the predicted soft label for sample $x_{\text{TP},0}$ under accuracy attack at $t$-th iteration as follows:

$$f(x_{\text{TP},0}^{t,\text{Acc}}) = f(x_{\text{TP},0}^{t-1,\text{Acc}} + \alpha \text{sign}(\nabla_{x_{\text{TP},0}^{t-1,\text{Acc}}} L_{\text{CE}}))$$

$$\approx f(x_{\text{TP},0}^{t-1,\text{Acc}}) + \alpha (\nabla_{x_{\text{TP},0}^{t-1,\text{Acc}}} f(x_{\text{TP},0}^{t-1,\text{Acc}}))^T \text{sign}(-\frac{1}{f(x_{\text{TP},0}^{t-1,\text{Acc}})} \nabla_{x_{\text{TP},0}^{t-1,\text{Acc}}} f(x_{\text{TP},0}^{t-1,\text{Acc}}))$$

$$= f(x_{\text{TP},0}^{t-1,\text{Acc}}) + \alpha (\nabla_x f(x_{\text{TP},0}^{t-1,\text{Acc}}))^T \text{sign}(\nabla_{x_{\text{TP},0}^{t-1,\text{Acc}}} L_{\text{CE}})$$

$$= f(x_{\text{TP},0}^{t-1,\text{Acc}}) - \alpha \|\nabla_{x_{\text{TP},0}^{t-1,\text{Acc}}} f(x_{\text{TP},0}^{t-1,\text{Acc}})\|_1$$

$$= f(x_{\text{TP},0}^{t-1,\text{Acc}}) - \delta_{\text{TP},0}^{t-1,\text{Fair}},$$

fwhere $\delta_{\text{TP},0}^{t,\text{Fair}} := \alpha \|\nabla_{x_{\text{TP},0}^{t-1,\text{Acc}}} f(x_{\text{TP},0}^{t-1,\text{Acc}})\|_1$ is the change of $x_{\text{TP},0}$'s predicted label under $\epsilon$-level fairness attack at $t$-th iteration since both are equivalent regarding $x_{\text{TP},0}$. This shows that disadvantaged TP samples that attains $\delta$-level robustness under $\epsilon$-level fairness attack also attains similar robustness w.r.t. accuracy attack.

For $x_{\text{TP},1}$, let $\delta_{\text{TP},1}^{t,\text{Acc}} := |f(x_{\text{TP},1}^{t,\text{Acc}}) - f(x_{\text{TP},1}^{t-1,\text{Acc}})|$, we have its change in predicted soft label under accuracy attack at $t$-th iteration as follows:

$$\delta(x_{\text{TP},1}^{t,\text{Acc}})$$

$$= |f(x_{\text{TP},1}^{t,\text{Acc}}) - f(x_{\text{TP},1}^{t-1,\text{Acc}})|$$

$$= |f(x_{\text{TP},1}^{t-1,\text{Acc}} + \alpha \text{sign}(\nabla_{x_{\text{TP},1}^{t-1,\text{Acc}}} L_{\text{CE}})) - f(x_{\text{TP},1}^{t-1,\text{Acc}})|$$

$$\approx \alpha (\nabla_{x_{\text{TP},1}^{t-1,\text{Acc}}} f(x_{\text{TP},1}^{t-1,\text{Acc}}))^T \text{sign}(\nabla_{x_{\text{TP},1}^{t-1,\text{Acc}}} L_{\text{CE}})$$

$$= \alpha \|\nabla_{x_{\text{TP},1}^{t-1,\text{Acc}}} f(x_{\text{TP},1}^{t-1,\text{Acc}})\|_1$$

$$\leq \delta_{\text{TP},0}^{t,\text{Fair}} + \sqrt{n} \alpha K d(x_{\text{TP},0}^{t-1,\text{Acc}}, x_{\text{TP},1}^{t-1,\text{Acc}}).$$

(16)

Taking the summation over all iterations, we have

$$\delta_{\text{TP},1}^{\text{Acc}} \leq \delta_{\text{TP},0}^{\text{Fair}} + \sum_{t=1}^{T} \sqrt{n}\alpha K d(x_{\text{TP},0}^{t-1,\text{Acc}}, x_{\text{TP},1}^{t-1,\text{Acc}}), \tag{17}$$

where $\delta_{\text{TP},0}^{\text{Fair}}$ is the change of predicted soft label of sample $x_{\text{TP},0}$ under $\epsilon$-level fairness attack. Since the inequality hold true for all $x_{\text{TP},0}$, we can further write (17) as

$$\delta_{\text{TP},1}^{\text{Acc}} \leq \min_{x_{\text{TP},0} \in \mathbb{S}_{10}} \delta_{\text{TP},0}^{\text{Fair}} + \sum_{t=1}^{T} \sqrt{n}\alpha K d(x_{\text{TP},0}^{t-1,\text{Acc}}, x_{\text{TP},1}^{t-1,\text{Acc}}).$$

And the lower bound $\delta_{\text{TP},1}^{\text{Acc}} \geq 0$ naturally holds true for samples under accuracy attack. This shows that for samples in the advantaged group, the change of predicted soft label under accuracy attack is lower-bounded by the fairness robustness of its neighbor sample(s) in the disadvantaged group up to an additive constant. For $f''$ under adversarial training w.r.t. fairness and $f$ under normal training, we have similar upper-bound except that we now have $\delta_{\text{TP},0}^{\text{Fair}} \geq \delta_{\text{TP},0}^{''\text{Fair}}$, which indicates that the adversarial classifier achieves tighter upper-bound than that of a normal classifier. $\square$

## I. Empirical verification of theoretical results

| $\epsilon$ | Method | $D_{\text{FN,male}}^{\text{Fair}}$ | $D_{\text{FN,female}}^{\text{Fair}}$ |
|------|--------|--------|--------|
| 0.1 | Baseline | 0.16±0.03 | 0.18±0.02 |
| 0.1 | Adversarial training (preprocessing) | **0.07±0.02** | **0.09±0.02** |
| 0.1 | Adversarial training (in-processing) | 0.07±0.02 | 0.11±0.02 |
| 0.1 | Adversarial training (post-processing) | 0.08±0.01 | 0.09±0.02 |
| 0.3 | Baseline | 0.23±0.02 | 0.26±0.03 |
| 0.3 | Adversarial training (preprocessing) | **0.09±0.02** | 0.11±0.02 |
| 0.3 | Adversarial training (in-processing) | 0.10±0.02 | 0.12±0.02 |
| 0.3 | Adversarial training (post-processing) | 0.10±0.02 | **0.09±0.01** |

*Table 4.* Change of cross-entropy loss for FN samples on CelebA dataset under fairness attacks with $\epsilon = 0.1$ and $\epsilon = 0.3$. Experiments are repeated three times.

We empirically validate the effectiveness of the upper-bounds stated in Theorem 5.3. Results on the change of cross-entropy loss for samples from different groups by baseline and by fair adversarial training under different perturbation levels are shown in Tab. 4. Under fair adversarial training, both advantaged and disadvantaged groups show improvements in $D^{\text{Fair}}$ compared with the baseline, which validates our theoretical results, that is, the alignment between fairness robustness and accuracy robustness.

Results of DP and EOd, as well as their theoretical bounds by Theorem 5.5 under varying levels of fairness attacks on CelebA dataset are shown in Fig. 5. While the theoretical bounds differ from the ground-truth values, they effectively capture the difference in the fairness robustness between the baseline and our method, validating the effectiveness of our analysis.

## J. Results of robustness against DP attack

We include the results of fair adversarial training in Fig. 6 to better distinguish between different fairness methods. Results of classifiers under DP attack on COMPAS, German and CelebA datasets are shown in Fig. 7.

## K. Extension to alternative adversarial training methods

We consider the following alternatives to PGD-based adversarial training, including Trades (Zhang et al., 2019) and Mart (Wang et al., 2019) for our fair adversarial training framework, and the results on robustness against fairness attack on CelebA dataset are shown in Fig. 8. Compared with vanilla adversarial training methods, our framework demonstrates remarkable improvement in fairness robustness, which validates that our proposed framework generalizes well to various adversarial training techniques.

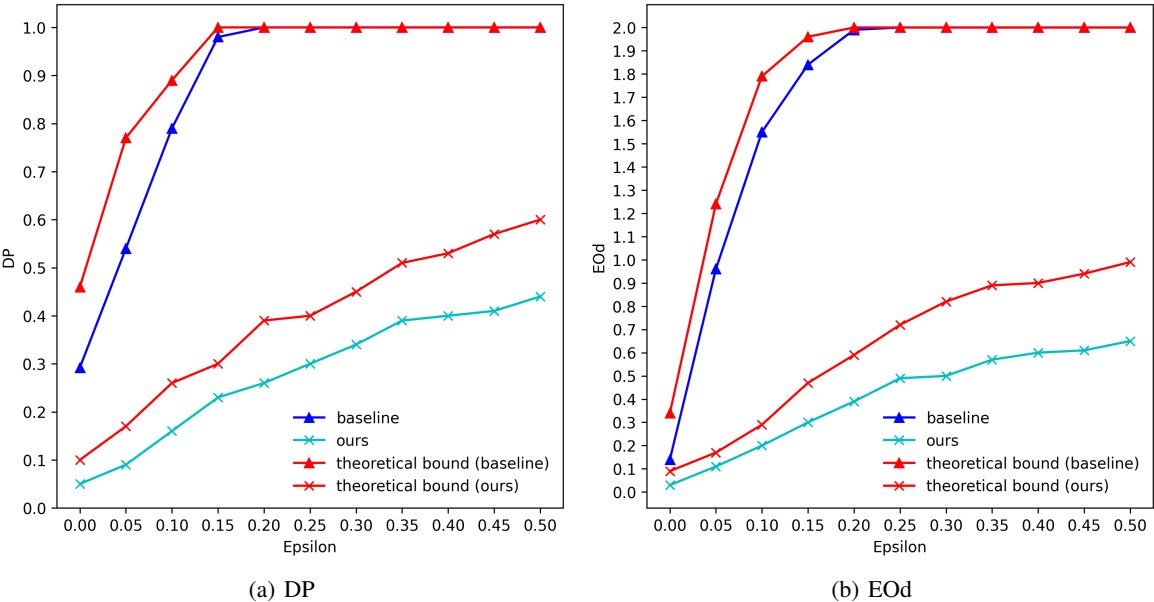

(a) DP            (b) EOd

*Figure 5.* Visualizations of theoretical bounds on the CelebA dataset under fairness attacks, where "ours" refers to adversarial training (preprocessing).

## L. Results of robustness against accuracy attack

We show the results on robustness against accuracy attack on COMPAS, German and CelebA datasets in Fig. 9-11.

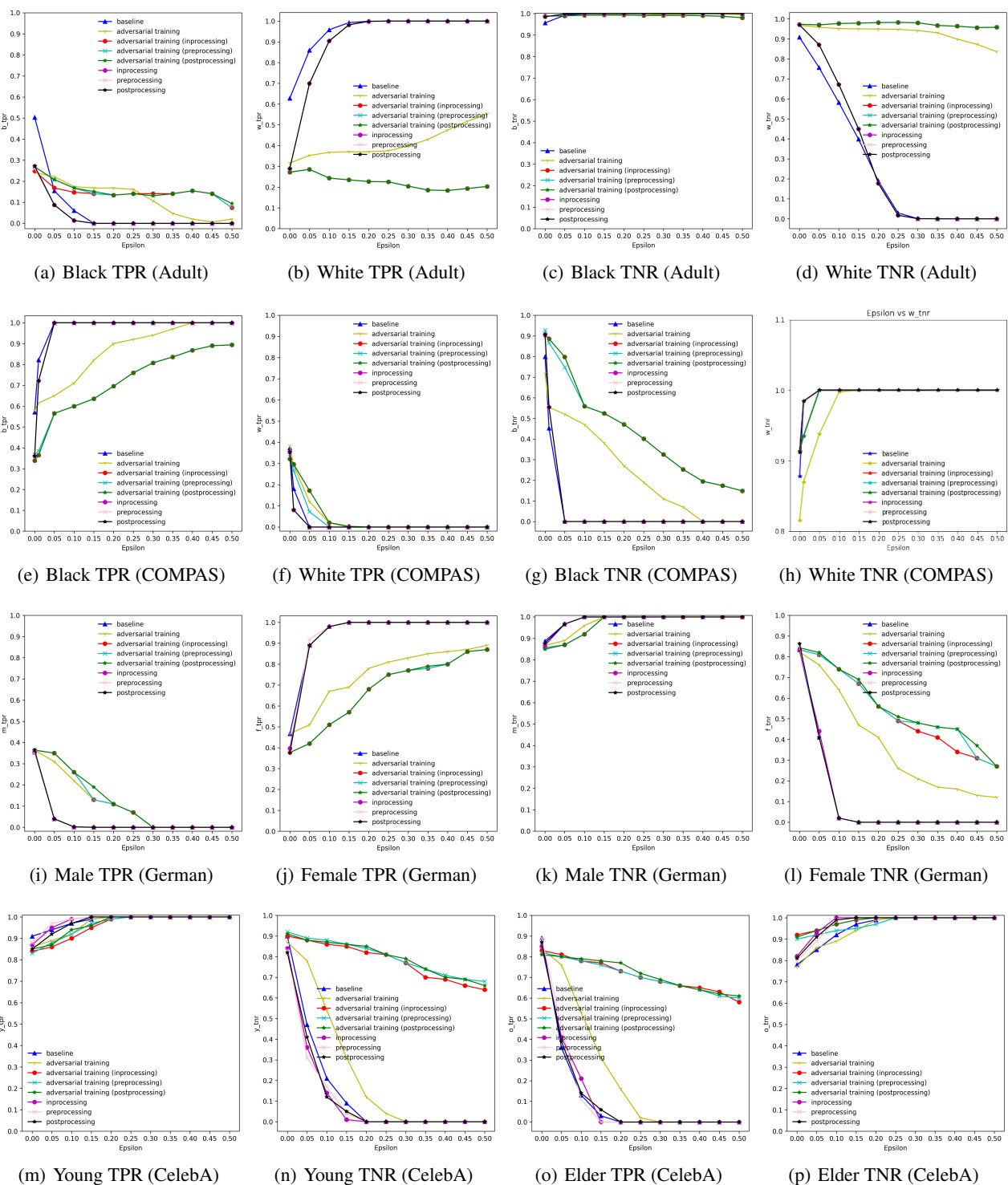

*Figure 6.* Change of true positive rate (TPR) and true negative rate (TNR) under fairness attacks on the four datasets.

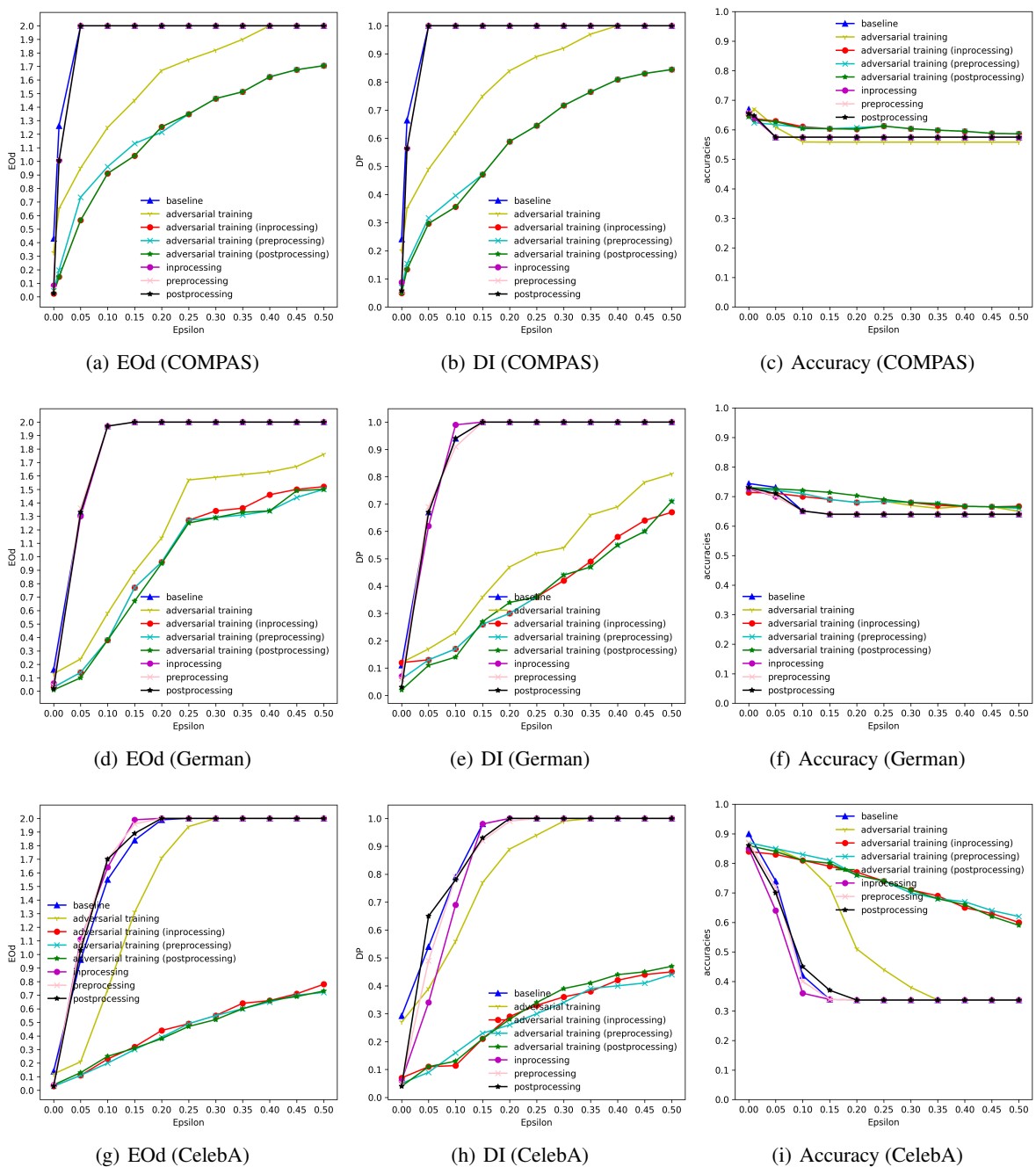

(a) EOd (COMPAS)   (b) DI (COMPAS)   (c) Accuracy (COMPAS)

(d) EOd (German)   (e) DI (German)   (f) Accuracy (German)

(g) EOd (CelebA)   (h) DI (CelebA)   (i) Accuracy (CelebA)

*Figure 7.* Change in accuracy, DI and EOd under fairness attacks on German dataset. Our adversarial training methods (preprocessing, in-processing, post-processing) obtain improved fairness (lower EOd and DI) and higher accuracy with significant margin.

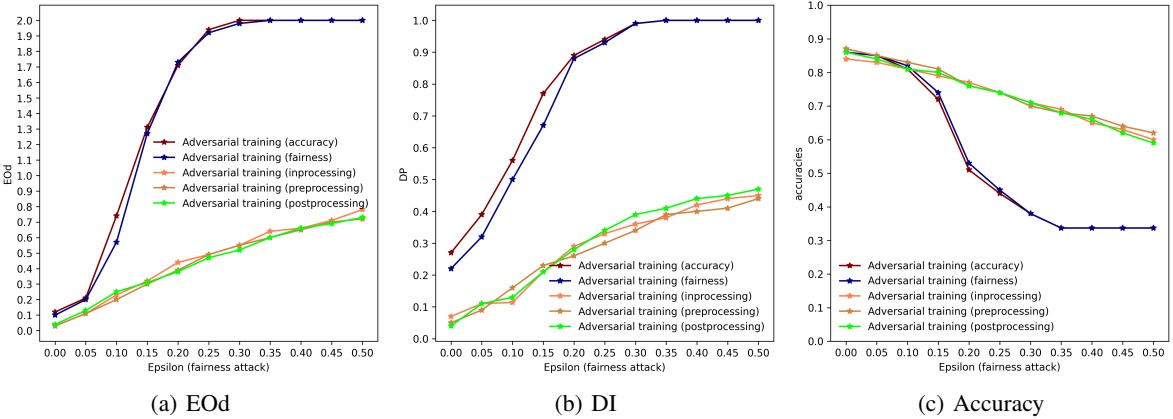

*Figure 8.* Experimental results on CelebA dataset under fairness attacks.

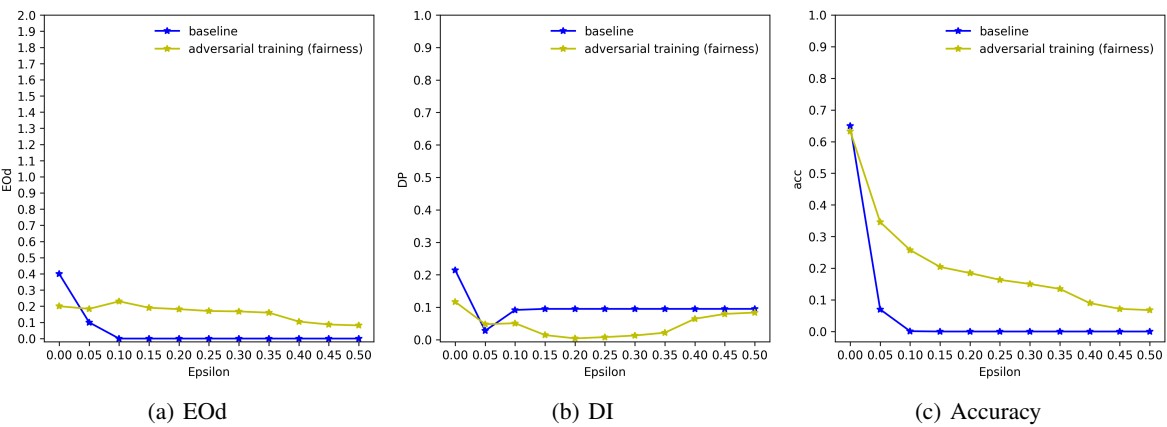

*Figure 9.* Results of a classifier adversarially trained w.r.t. DP. Change of accuracy, DP and EOd under accuracy attacks on COMPAS dataset.

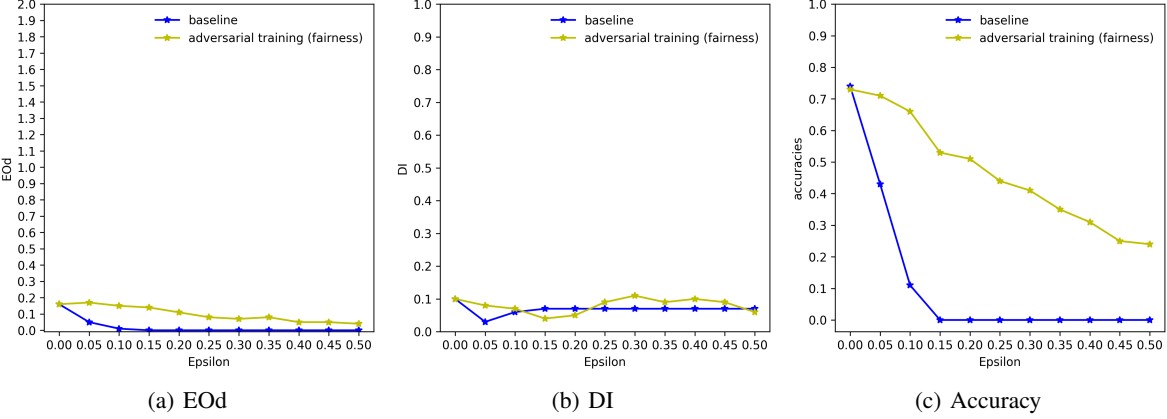

*Figure 10.* Results of a classifier adversarially trained w.r.t. DP. Change of accuracy, DP and EOd under accuracy attacks on German dataset.

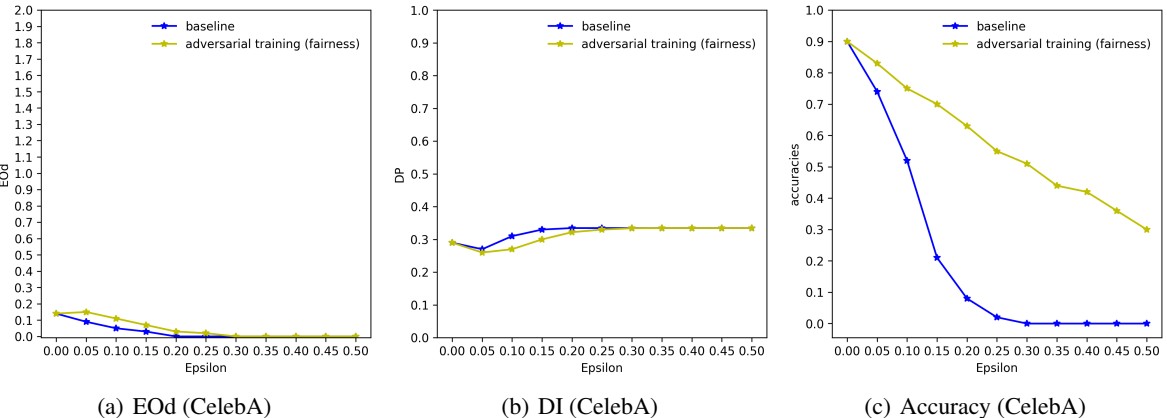

(a) EOd (CelebA)  (b) DI (CelebA)  (c) Accuracy (CelebA)

*Figure 11.* Results of a classifier adversarially trained w.r.t. DI. Change of accuracy, DP and EOd under accuracy attacks on CelebA dataset.

