# OpenReview forum: "On the Alignment between Fairness and Accuracy: from the Perspective of Adversarial Robustness"
_ICML.cc/2025/Conference — ICML 2025 poster_

### Official Review · Reviewer_qsrr · 2025-03-12

**Overall Recommendation:** 1

**Summary:**

This work theoretically discusses adversarial attacks against fairness, include the connection between adversarial attacks on fairness and those on accuracy, the connection between accuracy adversarial robustness and fairness  adversarial robustness.

**Claims And Evidence:**

The work makes several claims, but the valuable insights are limited. For example, Theorem 5.3 and Theorem 5.5 establish the relationship between attack robustness in terms of accuracy and fairness, leading to the conclusion that improving accuracy robustness and fairness in real-world applications can enhance fairness robustness. However, this is not a surprising result and lacks effective quantitative guidance for the community.

Most claims are supported by theoretical evidence; however, multiple assumptions are made to simplify the theorems, which limits the applicability of the results.

I also have concerns regarding the following claim:

The DP fairness attack aims to maximize the difference in the positive label rate between the advantaged and disadvantaged groups. However, in Section 4.1, it is simplistically analyzed as maximizing predictions in the advantaged group and minimizing them in the disadvantaged group. In real-world scenarios, it is also possible for the positive rate to increase in both groups, but with a greater increase in the advantaged group. Therefore, I believe that this overly simplified analysis may be problematic, especially since its validity serves as the foundation for the subsequent Corollary 4.2.

**Essential References Not Discussed:**

n/a

**Experimental Designs Or Analyses:**

The settings of the adversarial attack in the experiments are not introduced.

Some observations in Figure 3 do not align with the analysis. For example, when adversarial training w.r.t. fairness is introduced, accuracy robustness deteriorates rather than improving, which contradicts the analysis stating that 'accuracy robustness also benefits from adversarial training w.r.t. fairness.'

Additionally, I am confused about the significant performance differences in Figure 4 and Figure 2 under fairness attacks across the two datasets after adversarial training.

**Methods And Evaluation Criteria:**

I believe evaluating the tightness of the theoretical bounds through visualized experimental results would be more effective.

Additionally, there is a lack of detailed information regarding the setup of the adversarial attack scheme.

**Other Comments Or Suggestions:**

n/a

**Other Strengths And Weaknesses:**

n/a

**Questions For Authors:**

Could the authors provide a discussion on the insights from this work's results regarding the design of more effective adversarial attacks, such as those targeting only accuracy or fairness, as well as the corresponding defense strategies?

**Relation To Broader Scientific Literature:**

This work focuses on fairness-oriented adversarial attacks, an important topic. It aims to clarify the theoretical connections between adversarial attacks on fairness and accuracy in terms of attack effectiveness and adversarial robustness, making it a timely contribution. However, the technical innovations are limited, and the work does not provide insights based on its theoretical results for designing more effective adversarial attacks or corresponding defense strategies targeting specific performance aspects, only accuracy or only fairness.

Additionally, fairness metrics are diverse and sometimes conflicting. This work focuses on two specific forms, EOd and DP, but other metrics, such as the equal error rate, might lead to different conclusions. I would expect the findings of this work to be more generally applicable across a broader range of fairness metrics.

**Theoretical Claims:**

The proofs in the appendix seem correct, except that they require introducing assumptions on the foundation and boundary relaxations.

---

> ### Author Rebuttal · Authors · 2025-04-01
>
> We thank the reviewer for the detailed comments.
>
> [Theoretical Contribution and Insights]
>
> Our discussion is not focused on designing novel attack schemes, as this has been extensively covered in the existing literature. Instead, our goal is to identify efficient defense strategies against fairness attacks, which has not been adequately addressed in existing work.
>
> Our results contradict existing studies that suggest a trade-off between fairness and adversarial robustness. For example, [1] observes that in linear classifiers, a fairer model can be more vulnerable to accuracy attacks compared to a baseline model. Similarly, [2] empirically finds that reliance on sensitive information (i.e., exacerbating group-wise disparities) can enhance adversarial robustness. These observations suggest that adversarial training alone may be sufficient to improve fairness robustness. However, our analysis indicates that when designing defense strategies against fairness attacks, both static fairness and adversarial robustness shall be taken into account. This conclusion is supported by both our theoretical insights and empirical results under fairness attacks.
>
> [Applicability of Theoretical Results]
>
> As discussed in Section 5 of the main paper, the two assumptions we adopt align with those used in existing work and are introduced to facilitate our analysis. Although they may lead to slightly looser bounds, our analysis in Appendix I and the visualization results below validate the effectiveness of our theoretical framework in quantifying the alignment between fairness robustness and accuracy robustness. While the theoretical bounds differ from real-world values, they exhibit similar trends and preserve the relative ordering, where smaller upper-bounds correspond to smaller real-world values.
>
> [Formulation of the DP Attack]
>
> Since DP is defined by the difference in positive prediction rates, maximizing DP under a given perturbation level is most efficiently achieved by increasing one group's rate while decreasing the other's. In contrast, increasing the positive prediction rates for both groups does not align with the objective of maximizing disparity.
>
> [Visualization of Theoretical Bounds]
>
> Thank you for the suggestion. We show DP and EOd, as well as their theoretical bounds under varying levels of fairness attacks on CelebA dataset in the following link:
>
> https://drive.google.com/file/d/1nR1o4IHUOxFAhNc0AvLac2e9L1ehFYvD/view?usp=sharing
>
> We 'll include full results in the revised paper.
>
> [Setting of Adversarial Attacks]
>
> We refer to Sec. A of the appendix for the detailed setup.
>
> [Misalignment between Experimental Results and Analysis]
>
> We apologize for the confusion. The labels for "baseline" and "adversarial training (fairness)" in Fig. 3 are swapped. Specifically, "baseline" should be labeled as "adversarial training (fairness)," and "adversarial training (fairness)" should be labeled as "baseline." We 'll correct the mislabeling in the revised paper.
>
> [Performance Differences in Figure 2 and 4]
>
> The performance differences between the baseline, vanilla adversarial training and our method validate our analysis in Theorem 5.5 of the main paper. Our analysis demonstrates that to achieve smaller changes in group-wise disparities under a fairness attack, it is essential to consider both static fairness and accuracy robustness; focusing on only one does not guarantee low fairness violations. In contrast, our method jointly addresses both aspects, resulting in lower fairness violations compared with other methods.
>
> [Generalization to Other Metrics]
>
> Our formulation can be generalized to alternative fairness notions as it aims to maximize disparities in predictions between groups. The two metrics we focus on have been shown to be conflicting under varying base rates [3], but by maximizing DP, we also maximize EOd. We show results of two alternative metrics, predictive equality (PE) [3] and positive class balance (PCB) [4] on CelebA dataset under the perturbation level $\epsilon=0.15$ in the following link:
>
> https://drive.google.com/file/d/1CS41yfL4YwBwYakPQVQVtxDqoLX-e4xn/view?usp=sharing
>
> The DP attack effectively maximizes PE and PCB on the baseline, while our method maintains lower values for both metrics, validating the generalizability of our formulation and our defense framework.
>
> [1] Tran, Cuong, et al. "Fairness increases adversarial vulnerability." arXiv preprint arXiv:2211.11835 (2022).
>
> [2] Moayeri, Mazda, Kiarash Banihashem, and Soheil Feizi. "Explicit tradeoffs between adversarial and natural distributional robustness." Advances in Neural Information Processing Systems 35 (2022): 38761-38774.
>
> [3] Chouldechova, Alexandra. "Fair prediction with disparate impact: A study of bias in recidivism prediction instruments." Big data 5.2 (2017): 153-163.
>
> [4] Kleinberg, Jon, Sendhil Mullainathan, and Manish Raghavan. "Inherent trade-offs in the fair determination of risk scores." arXiv preprint arXiv:1609.05807 (2016).

---

### Official Review · Reviewer_5adC · 2025-03-12

**Overall Recommendation:** 4

**Summary:**

The paper "On the Alignment between Fairness and Accuracy: from the Perspective of Adversarial Robustness" explores the connections between adversarial training for fairness and accuracy objectives. The authors demonstrate the potential synergy between these two robustness goals. After a theoretical analysis of their alignment - showing that fairness robustness can benefit from adversarial accuracy training under a relaxed fairness control - the authors present experiments that support their claims. The results indicate that even without specific fairness fine-tuning against fairness attacks, a model trained adversarially for accuracy robustness retains reasonable fairness properties. Furthermore, improving fairness by controlling the adversarial dataset during preprocessing, optimizing under a relaxed fairness constraint (in-processing), or applying fairness post-processing to the outputs is shown to further enhance fairness robustness, in line with theoretical predictions. I believe this paper could be valuable to the community for designing models that are robust with respect to both criteria.

**Claims And Evidence:**

Yes. Experiments look convincing and well follow theoretical claims. While I did not fully check all proofs, the theoretical analysis looks well sounded.

My only concern regarding the experiments is the use of fairness attacks for accuracy robustness (Figure 3). While the authors claim in the accompanying text that fairness adversarial training significantly improves accuracy robustness, I observe the opposite in the reported curves. Maybe the legends have been swapped?

**Essential References Not Discussed:**

.

**Experimental Designs Or Analyses:**

.

**Methods And Evaluation Criteria:**

.

**Other Comments Or Suggestions:**

.

**Other Strengths And Weaknesses:**

Strengths :
- Interesting theoretical of fairness and accuracy robustness alignement
- Experiments that well match the theory

Weaknesses :
- Maybe confined to binary classification and binary sensitives
- Doesn't fully follow classical definition of fairness metrics for binary classification, that usually considers the final binary decision rather than the class probablity (this is what justifies adversarial fair training rather than relaxed constraints on differentiable statistic, see for instance [2] that discusses that point in section A.2.2, starting from the approach in [1]).


[1]  Brian Hu Zhang, Blake Lemoine, and Margaret Mitchell. Mitigating unwanted biases with adversarial learning. In Proceedings of the 2018 AAAI/ACM Conference on AI, Ethics, and Society, pp. 335–340, 2018.

[2] Grari, Vincent, et al. "On the Fairness ROAD: Robust Optimization for Adversarial Debiasing." The Twelfth International Conference on Learning Representations. 2023.

**Questions For Authors:**

- The paper looks limited to binary classification and binary sensitives. How could the work be extended behind that restricted setting ?
- Theoretical analysis considers only fully succesful attacks (see for instance the proof of corollary 4.1) where f(x)=1 for all individuals from the privileged group. But doesn't is correspond to a kind of shift of analysis, since in practice this is not the case, at least for some reasonnable bounds for the attack norms ?
- As mentionned above, fairness metrics considered in the paper are relaxed versions of original ones. What would be the results with 0-1 fairness metrics ? How could this limiting setting be overcome ? I feel reporting fairness metrics based on outcome probabilities is not enough since the unprivileged group could can have better probas in average (for instance 0.49 for everyone in the group), while the other group getting probas close to 0.5 also (for instance 0.51 for everyone), reporting a quite fair model, while being strongly unfair when outcomes are thresholded.
- In algo 2, the relaxed fairness constraint looks a bit difficult to be optimized as it. are the statictics performed on the minibatch only ? Is it enough ?


Presentation remarks  :
- I feel that the paper should be more self-contained regarding the fairness mitigation approaches that are used in the algorithms. Rather than only saying "reweighting data by Yu et al" in algo 1, or "post-process by Jang et al." in algo 3, it would be nice if authors could give even a minimal rationnale of what these are.
- The projection operator is not well defined in section 3.1. And it is confusing, as it looks as a product at a first glance. Also, is x+S correct, I feel this formalism is misleading (I would say S(x^t) for a given ball around x^t of something like that)
- Directly above (2), the text mentions L_{DI}, while (2) gives L_{DP}. same line 204 for the gradient.
- line 237 second columns. whe have $\delta^{Fair}$ defined with $x^{DP}$. Correct ?
- Assumption 5.2 doesn't look fully clear to me. What do you mean for a bound of a distribution ?
- The text in theroem 5.3 is somehow also confusing to me, as it mentions the difference of fairness between false negatives of both groups that would be bounded, while tha LHS of the inequality only gives the robustness of the group 1.
-

**Relation To Broader Scientific Literature:**

.

**Theoretical Claims:**

.

---

> ### Author Rebuttal · Authors · 2025-04-01
>
> We thank the reviewer for the detailed comments.
>
> [W1: Extension to Non-Binary Classification]
>
> Our method can be readily extended to multi-class scenarios by simply replacing $L_{\text{CE}}$ and the fairness constraint $L$ with their respective multi-class formulations. We show results in the following table on Drug dataset [1] to validate the extension, where the perturbation level is chosen as $0.1$:
>
> |Method   | Accuracy | DP | EOd |
> | --------- | -------- | -------- | -------- |
> | Baseline| 0.41   |0.96    |1.97    |
> | Advcersarial Training| 0.63   |0.49    |1.03    |
> | Advcersarial Training (preprocessing)| 0.64   |0.15    |0.22   |
> | Advcersarial Training (in-processing)| 0.63   |0.17    |0.23   |
> | Advcersarial Training (post-processing)| 0.64   |0.18    |0.25   |
>
> Our method (last three rows) achieves remarkably better performance under the fairness attack, validating the extensibility to non-binary tasks.
>
> [W2: Theoretical Analysis]
>
> We may clarify that our analysis is not limited to fully successful attacks. Under a successful DP attack, we have $f(x) < 0.5$ for samples in the disadvantaged group $\mathbb{S}\_{.0}$, and $f(x) \geq 0.5$ for samples in the advantaged group $\mathbb{S}\_{.1}$. Consequently, we have
> \begin{align*}
> \text{EOd} = &\\left|\\frac{\\sum_{x \in \mathbb{S}\_{10}} \mathbb{1}[f(x) \geq 0.5]}{|\mathbb{S}\_{10}|} - \sum_{x \in \mathbb{S}\_{11}} \frac{\mathbb{1}[f(x) \geq 0.5]}{|\mathbb{S}\_{11}|}\right|
> \\
> &+ \left|\sum_{x \in \mathbb{S}\_{00}} \frac{\mathbb{1}[f(x) < 0.5]}{|\mathbb{S}\_{00}|} - \sum_{x \in \mathbb{S}\_{01}} \frac{\mathbb{1}[f(x) < 0.5]}{|\mathbb{S}\_{01}|}\right|
> \\
> & =2,
> \end{align*}
>
> which indicates that a successful DP attack implies a successful EOd attack. However, the converse does not hold true, as discussed in the counterexample of Appendix D. Furthermore, our analysis in Sec. 5 of the main paper on the alignment between fairness robustness and accuracy robustness does not rely on additional assumptions about the attacks. The discussion on fully successful attacks is included solely to illustrate this relationship.
>
> [W3: Fairness Metrics]
>
> We are sorry for the confusion. The results we reported are calculated based on 0-1 fairness metrics, rather than the relaxed version. We 'll clarify the choice in the revised paper.
>
> [W4: Optimization of Algorithm 2]
>
> The relaxed fairness constraint is optimized over each mini-batch. Since the mini-batches are randomly constructed from the training data, as long as the batch size is not excessively small, $L_{\text{DI}}$ can be reliably estimated and optimized using only mini-batches.
>
> [R1: Algorithm Presentation]
>
> Thank you for the suggestion. We 'll include the details of the fairness interventions in Algorithm 1 and 3 in the revised paper. The preprocessing method [2] reweighs each training sample to balance the class distributions within each sensitive group, while the post-processing method by [3] adjusts the decision threshold for each group based on approximated logit distributions.
>
> [R2: Projection Operator]
>
> Our formulation primarily follows the conventional formalism [4]. We 'll include more explanations in the revised paper to avoid confusion.
>
> [R3&4: Notations]
>
> We are sorry for the confusion. $L\_{\text{DI}}$ should be $L\_{\text{DP}}$. Since $\delta^{\text{fair}}\_{\text{sub,a}}$ is defined as the change in $L\_{\text{CE}}$ before and after the fairness attack, we slightly abuse the notation here, as we focus on the DP attack as the fairness attack.
>
> [R5: Assumption 5.2]
>
> By "bound of a distribution" we refer to the supremum of function values of a distribution, i.e.,
> $$
> \max_{x} p(x),
> $$
> where $p$ is the probability density function (PDF) of the distribution.
>
> [R6: Clarification of Theroem 5.3]
>
> As discussed in Sec. 5 (line 272-274) of the main paper, the fairness robustness of $x_{\text{FN},0}$ naturally aligns with the accuracy robustness of $x_{\text{FN},0}$ since the fairness attacks and accuracy attacks are identical regarding $x_{\text{FN},0}$. Consequently, $x_{\text{FN},0}$ is naturallly bounded under adversarial training. Therefore, our discussion in Theroem 5.3 focuses on $x_{\text{FN},1}$, where the fairness attack diverges from the accuracy attack.
>
> [1] Fehrman, E., Egan, V., & Mirkes, E. (2015). Drug Consumption (Quantified) [Dataset]. UCI Machine Learning Repository. https://doi.org/10.24432/C5TC7S.
>
> [2] Yu, Zhe, Joymallya Chakraborty, and Tim Menzies. "FairBalance: How to Achieve Equalized Odds With Data Pre-processing." IEEE Transactions on Software Engineering (2024).
>
> [3] Jang, Taeuk, Pengyi Shi, and Xiaoqian Wang. "Group-aware threshold adaptation for fair classification." Proceedings of the AAAI Conference on Artificial Intelligence. Vol. 36. No. 6. 2022.
>
> [4] Madry, Aleksander, et al. "Towards deep learning models resistant to adversarial attacks." arXiv preprint arXiv:1706.06083 (2017).

---

### Official Review · Reviewer_Mgij · 2025-03-13

**Overall Recommendation:** 2

**Summary:**

The authors introduce a cohesive framework for adversarial training that can be adapted to multiple definitions of group fairness.  The general idea is to formulate a certain objective function that captures the loss and then to perturb the input in a direction given by the gradient so as to increase the loss. Then, the paper experimentally shows how adversarially trained models achieve robustness to both accuracy attacks and fairness attacks.

**Claims And Evidence:**

The claims within this paper are well supported roughly.

**Essential References Not Discussed:**

There are no essential references missing.

**Experimental Designs Or Analyses:**

The experimental design is valid and well-organized.

**Methods And Evaluation Criteria:**

The evaluation criteria, such as datasets (e.g. CelebA) are popular and sufficient for assessing performance.

**Other Comments Or Suggestions:**

- The writing should be polished. There are same typos shown in many places, for example, "the the fairness attack".

- The mathematical notations are quite intricate. I strongly recommend that the authors present them in a more accessible format, such as tables, for easier reading.

**Other Strengths And Weaknesses:**

**Strengths**

- The paper extend the typical adversarial attacks problem towards accuracy and then explores the fairness problem under adversarial attacks.
- The fairness robustness under adversarial attacks is worthy investigating.
- The empirical results provide enough convincing evidences.

**Weaknesses**

- Given the vague connections between features and sensitive attributes, it seems to be hard to understand the perturbations on these sensitive attributes.

**Questions For Authors:**

I don't have any specific questions

**Relation To Broader Scientific Literature:**

This paper investigates the fairness problem in the context of adversarial attacks, which could be potential useful for future research.

**Theoretical Claims:**

Although I have not examined the proofs in detail, the theoretical claims appear to be generally correct.

---

> ### Author Rebuttal · Authors · 2025-04-01
>
> We thank the reviewer for the detailed comments. We will carefully refine the writing and include tables of notations in the revised paper to enhance readability. We include a short list of notations in the following table:
>
> |Notation  | Meaning|
> | --------- | ------------|
> |$x_{\text{sub},a}^{t, \text{obj}}$|Adversarial sample(s) generated from the clean subgroup {sub, a} at t-th iteration, targeting obj (Acc, DP, EOd)
> |$\mathbb{S}_{\text{sub},a}^{t, \text{obj}}$|The set of adversarial samples at t-th iteration from the subgroup {sub, a}|
> |$x_{\text{sub},a}^{\text{obj}}$|Adversarial sample(s) obtained after the adversarial attack targeting obj|
> |$p_{\text{sub},a}^{\text{obj}}$|The distribution of predicted soft labels in the clean subgroup {sub, a} after the adversarial attack targeting obj|
> |$x_{\text{sub},a}$ |Clean samples from the subgroup {sub, a}|
> |$p_{\text{sub},a}$|The distribution of predicted soft labels in the clean subgroup {sub, a} before the adversarial attack|
>
> [W1: Clarification of Adversarial Perturbation]
>
> Regarding the fairness attack, while the adversarial objective is formulated at the group level, the perturbations are applied to individual input samples rather than directly modifying sensitive attributes. Similar to how labels guide perturbation directions in the accuracy attack, sensitive attributes are used solely to determine the perturbation directions in the fairness attack. Specifically, the fairness attack shifts samples from the advantaged group toward the subspace of positive predictions, while pushing samples from the disadvantaged group toward the subspace of negative predictions.

---

### Official Review · Reviewer_yumK · 2025-03-18

**Overall Recommendation:** 3

**Summary:**

The paper analyzes adversarial attacks and robustness with respect to both fairness and accuracy. The authors prove theoretically the equivalence of adversarial attacks against different fairness notions, like DP and EOD. The theoretical analysis also shows the connections between attacks targeting accuracy and those targeting fairness. In this sense, improvements in robust accuracy have a positive impact on robust fairness and vice versa. The authors also propose a fair adversarial training strategy which integrates adversarial training with fairness constraints to enhance both fairness and accuracy robustness. The experimental evaluation using four common benchmarks in the fairness research literature corroborates the theoretical results and the benefits of the proposed adversarial training strategy.

**Claims And Evidence:**

Yes, the claims made in the submission are supported by convincing evidence, both theoretical and empirical. The assumptions for the theoretical analysis are reasonable and common in the scope of the work.

Perhaps, it would be appropriate to include a discussion on the computational complexity of the proposed adversarial training approach compared to traditional approaches focusing just on robust accuracy.

**Essential References Not Discussed:**

None that I am aware.

**Experimental Designs Or Analyses:**

The experimental analysis is reasonable and serves to support the theoretical claims in the paper. The datasets and the models selected for the experiments are adequate and commonly used in the research literature in fairness.

**Methods And Evaluation Criteria:**

The scope of the work is well presented and reasonable. The proposed method seems sound, and the evaluation criteria is enough to support the theoretical claims made in the paper.

**Other Comments Or Suggestions:**

See the comments above.

**Other Strengths And Weaknesses:**

Strengths:
+ The paper provides a nice contribution to the fairness literature, especially in adversarial contexts.
+ The authors strived to provide a solid theoretical foundation supported by a reasonable empirical validation.

Weaknesses:
+ The paper would benefit from a clearer threat model, before the theoretical analysis is presented.
+ The paper does not address properly the computational trade-offs and discuss how this framework could be used in practical scenarios.

**Questions For Authors:**

+ Could the authors provide some insights on the computational trade-offs and the practicality of the proposed approach in real scenarios?

**Relation To Broader Scientific Literature:**

Fairness has been an aspect somewhat overlooked in the research literature in adversarial machine learning. Perhaps, this aspect has been more considered in the context of poisoning attacks, e.g., (Solans et al., 2020; Mehrabi et al., 2021b) among others. However, in the context of adversarial examples training has focused on some implications and difficulties of applying adversarial training when considering fairness. In this sense, as mentioned by the authors, some works like (Nanda et al., 2021; Xu et al., 2021; Ma et al., 2022) state that adversarial training without proper regularization leads to class-wise disparities in accuracy and robustness. To my best knowledge, this is the first work proposing a more general framework of attacks and defenses considering group fairness.

**Theoretical Claims:**

I quickly schemed through the different proofs included in the appendix. From this, the theoretical claims seem reasonable to me, although I did not have the chance to check everything thoroughly.

---

> ### Author Rebuttal · Authors · 2025-04-01
>
> We thank the reviewer for the detailed comments.
>
> [W1: Clarification of Threat Model]
>
> In our threat model, we assume that the adversarial has full access to the parameters of the target model. The adversarial manipulation is performed at the input level, subject to a maximum perturbation level $\epsilon$. The fairness attacks aim to maximize group-wise disparities on the testing data, quantified by metrics including Demographic Parity (DP) and Equalized Odds (EOd), while the accuracy attacks are designed to maximize the classification error on the testing data. We 'll include the description in the revised paper.
>
> [W2: Computational Trade-Off]
>
> We include the training time of our proposed framework relative to vanilla adversarial training in the following table, where the fairness intervention is chosen as in-processing [1]:
>
> |Dataset   | Adult | German | COMPAS | CelebA |
> | --------- | -------- | -------- | -------- | -------- |
> | Time| 1.32    |1.27    |1.22    |1.24    |
>
> Our method leads to relatively small increase in the training time compared with vanilla adversarial training, and therefore leads to a reasonable computational trade-off.
>
> [W3: Practical Applicability]
>
> As shown in Eq. 6, our method does not require additional assumptions about the model architecture, fairness interventions, or adversarial training techniques. Consequently, it can be seamlessly integrated into the training process of fair models by simply replacing clean samples with adversarial ones.
>
> [1] Wang, Jialu, Xin Eric Wang, and Yang Liu. "Understanding instance-level impact of fairness constraints." International Conference on Machine Learning. PMLR, 2022.

---

> > ### Comment · Reviewer_yumK · 2025-04-05
> >
> > Thank you very much for addressing my comments. I think that the paper would benefit from a clearer threat model at the beginning of the paper to help the readers understand better the problem and the assumptions made on the attacker. On the other hand, the extension to non-binary classification problems (as suggested by reviewer 5adC) with the results showed by the authors in the rebuttal could be a nice addition to the paper as well. After reading the rebuttal and the other reviewers' comments I'm keeping my positive score.

---

> > > ### Author Response · Authors · 2025-04-05
> > >
> > > Thank you very much for your recognition and valuable feedbacks of our work! We are glad that we have addressed your concerns.

---

### Decision · Program_Chairs · 2025-05-01

**Decision:**

Accept (poster)

**Comment:**

The paper proposes an adversarial attack on model fairness. While the reviewers provide strongly diverging scores, they agree that the proposed research question is novel and interesting. Major weaknesses are seen in the paper not being self-contained, mathematical notation not being clear, simplifications not always being motivated. After the rebuttal, the AC sees most of these concerns addressed. The amount of editorial changes to the manuscript that are promised by the authors makes the paper a borderline case, but the changes, while necessary, will be easy to implement. The authors will have to improve the manuscript by incorporating their responses, in particular to reviewers 5adC, yumK and Mgij into the final manuscript.